# Stateful Strategic Regression

**Keegan Harris**
School of Computer Science
Carnegie Mellon University
Pittsburgh, PA 15213
keeganh@cmu.edu

**Hoda Heidari**
School of Computer Science
Carnegie Mellon University
Pittsburgh, PA 15213
hheidari@cmu.edu

**Zhiwei Steven Wu**
School of Computer Science
Carnegie Mellon University
Pittsburgh, PA 15213
zstevenwu@cmu.edu

## Abstract

Automated decision-making tools increasingly assess individuals to determine if they qualify for high-stakes opportunities. A recent line of research investigates how strategic agents may respond to such scoring tools to receive favorable assessments. While prior work has focused on the *short-term* strategic interactions between a decision-making institution (modeled as a principal) and individual decision-subjects (modeled as agents), we investigate interactions spanning *multiple time-steps*. In particular, we consider settings in which the agent's effort investment today can accumulate over time in the form of an internal *state*—impacting both his future rewards and that of the principal. We characterize the Stackelberg equilibrium of the resulting game and provide novel algorithms for computing it. Our analysis reveals several intriguing insights about the role of multiple interactions in shaping the game's outcome: First, we establish that in our stateful setting, the class of all linear assessment policies remains as powerful as the larger class of all monotonic assessment policies. While recovering the principal's optimal policy requires solving a non-convex optimization problem, we provide polynomial-time algorithms for recovering both the principal and agent's optimal policies under common assumptions about the process by which effort investments convert to observable features. Most importantly, we show that with multiple rounds of interaction at her disposal, the principal is more effective at incentivizing the agent to accumulate effort in her desired direction. Our work addresses several critical gaps in the growing literature on the societal impacts of automated decision-making—by focusing on *longer time horizons* and accounting for the *compounding* nature of decisions individuals receive over time.

## 1 Introduction

Automated decision-making tools increasingly assess individuals to determine whether they qualify for life-altering opportunities in domains such as lending [27], higher education [32], employment [41], and beyond. These assessment tools have been widely criticized for the blatant disparities they produce through their scores [43, 3]. This overwhelming body of evidence has led to a remarkably active area of research into understanding the societal implications of algorithmic/data-driven automation. Much of the existing work on the topic has focused on the *immediate* or *short-term* societal effects of automated decision-making. (For example, a thriving line of work in Machine Learning (ML) addresses the unfairness that arises when ML predictions inform high-stakes decisions [18, 22, 31, 8, 1, 16, 11] by defining it as a form of predictive disparity, e.g., inequality in false-positive rates [22, 3] across social groups.) With the exception of several noteworthy recent articles (which we discuss shortly), prior work has largely ignored the *processes* through which algorithmic decision-making systems can *induce, perpetuate, or amplify* undesirable choices and behaviors.

35th Conference on Neural Information Processing Systems (NeurIPS 2021).

Our work takes a *long-term perspective* toward modeling the interactions between individual decision subjects and algorithmic assessment tools. We are motivated by two key observations: First, algorithmic assessment tools often provide predictions about the *latent* qualities of interest (e.g., creditworthiness, mastery of course material, or job productivity) by relying on *imperfect* but *observable* proxy attributes that can be directly evaluated about the subject (e.g., past financial transactions, course grades, peer evaluation letters). Moreover, their design ignores the *compounding* nature of advantages/disadvantages individual subjects accumulate over time in pursuit of receiving favorable assessments (e.g., debt, knowledge, job-related skills). To address how individuals *respond* to decisions made about them through modifying their observable characteristics, a growing line of work has recently initiated the study of the *strategic* interactions between decision-makers and decision-subjects (see, e.g., [15, 26, 36, 30, 21]). This existing work has focused mainly on the *short-term* implications of strategic interactions with algorithmic assessment tools—e.g., by modeling it as a *single round* of interaction between a principal (the decision-maker) and agents (the decision-subjects) [30]. In addition, existing work that studies interactions over time assumes that agents are myopic in responding to the decision-maker's policy [4, 42, 38, 15]. We expand the line of inquiry to *multiple rounds* of interactions, accounting for the impact of actions today on the outcomes players can attain tomorrow.

**Our multi-round model of principal-agent interactions.** We take the model proposed by Kleinberg and Raghavan [30] as our starting point. In Kleinberg and Raghavan's formulation, a principal interacts with an agent *once*, where the interaction takes the form of a Stackelberg game. The agent receives a score $y = f(\boldsymbol{\theta}, \mathbf{o})$, in which $\boldsymbol{\theta}$ is the principal's choice of assessment parameters, and $\mathbf{o}$ is the agent's observable characteristics. The score is used to determine the agent's merit with respect to the quality the principal is trying to assess. (As concrete examples, $y$ could correspond to the grade a student receives for a class, or the FICO credit score of a loan applicant.) The principal moves first, publicly announcing her assessment rule $\boldsymbol{\theta}$ used to evaluate the agent. The agent then best responds to this assessment rule by deciding how to invest a *fixed* amount of effort into producing a set of observable features $\mathbf{o}$ that maximize his score $y$. Kleinberg and Raghavan characterize the assessment rules that can incentivize the agent to invest in specific types of effort (e.g., those that lead to real *improvements* in the quality of interest as opposed to *gaming* the system). We generalize the above setting to $T > 1$ rounds of interactions between the principal and the agent and allow for the possibility of certain effort types rolling over from one step to the next. Our key finding is that longer time horizon provides the principal additional latitude in the range of effort sequences she can incentivize the agent to produce. To build intuition as to why repeated interactions lead to the expansion of incentivizable efforts, consider the following stylized example:

**Example 1.1.** Consider the classroom example of Kleinberg and Raghavan where a teacher (modeled as a principal) assigns a student (modeled as an agent) an overall grade $y$ based on his observable features; in this case test and homework score. Assume that the teacher chooses an assessment rule and assigns a score $y = \theta_{TE} TE + \theta_{HW} HW$, where $TE$ is the student's test score $HW$ is his homework score, and $\theta_T, \theta_{HW} \in \mathbb{R}$ are the weight of each score in the student's overall grade. The student can invest effort into any of three activities: copying answers on the test, studying, and looking up homework answers online. In a one-round setting where the teacher only evaluates the student once, the student may be more inclined to copy answers on the test or look up homework answers online, since these actions immediately improve the score with relatively lower efforts. However, in a multiple-round setting, these two actions do not improve the student's knowledge (which impacts the student's future grades as well), and so these efforts do not carry over to future time steps. When there are multiple rounds of interaction, the student will be incentivized to invest effort into studying, as knowledge accumulation over time takes less effort in the long-run compared to cheating every time. We revisit this example in further detail in Appendix A.

**Summary of our findings and techniques.** We formalize settings in which the agent's effort investment today can *accumulate* over time in the form of an internal *state*—impacting both his future rewards and that of the principal. We characterize the Stackelberg equilibrium of the resulting game and provide novel algorithmic techniques for computing it. We begin by establishing that for the principal, the class of all *linear* assessment policies remains as powerful as the larger class of all *monotonic* assessment policies. In particular, we prove that if there exists an assessment policy that can incentivize the agent to produce a particular sequence of effort profiles, there also exists a linear assessment policy which can incentivize the exact same effort sequence.

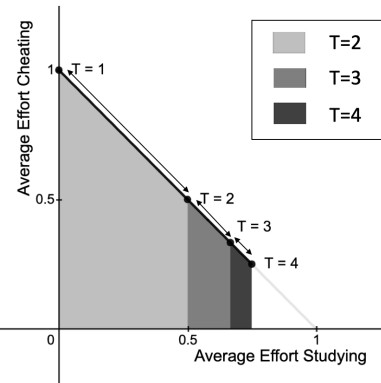

Figure 1: Average effort spent studying vs. average effort spent cheating over time for the example in Appendix A. The line $x + y = 1$ represents the set of all possible Pareto optimal average effort profiles. The shaded region under the line represents the set of average effort profiles which can be incentivized with a certain time horizon. Darker shades represent longer time horizons. In the case where $T = 1$, it is not possible to incentivize the agent to spend any effort studying. Arrows are used to demonstrate the additional set of Pareto optimal average effort profiles that can be incentivized with each time horizon. As the time horizon increases, it becomes possible to incentivize a wider range of effort profiles.

We then study the equilibrium computation problem, which in general involves optimizing *non-convex* objectives. Despite the initial non-convexity, we observe that when the problem is written as a function of the agent's incentivized efforts, the principal's non-convex objective becomes convex. Moreover, under a common assumption on agent's conversion mapping from efforts to observable features, the set of incentivizable effort policies is also convex. Given this structure, we provide a polynomial-time algorithm that directly optimizes the principal's objective over the set of incentivizable effort policies, which subsequently recovers agent's and principal's equilibrium strategies. Even though prior work [39, 40] has also taken this approach for solving other classes of non-convex Stackelberg games, our work has to overcome an additional challenge that the agent's set of incentivizable efforts is not known a-priori. We resolve this challenge by providing a *membership oracle* (that determines whether a sequence of agent efforts can be incentivized by any assessment policy), which allows us to leverage the convex optimization method due to Kalai and Vempala [28].

Our analysis reveals several intriguing insights about the role of repeated interactions in shaping the long-term outcomes of decision-makers and decision subjects: For example, we observe that with multiple rounds of assessments, both parties can be better off employing *dynamic/time-sensitive* strategies as opposed to *static/myopic* ones. Crucially, perhaps our most significant finding is that by considering the effects of multiple time-steps, the principal is significantly more effective at incentivizing the agent to accumulate effort in her desired direction (as demonstrated in Figure 1 for a stylized teacher-student example). In conclusion, our work addresses two critical gaps in the growing literature on the societal impacts of automated decision-making–by focusing on *longer time horizons* and accounting for the *compounding* nature of decisions individuals receive over time.

## 1.1 Related work

A growing line of work at the intersection of computer science and social sciences investigates the impacts of algorithmic decision-making models on people (see, e.g., [25, 44, 34, 15]). As we outline below, significant attention has been devoted to settings in which decision-subjects are strategic and respond to the decision-maker's choice of assessment rules. Liu et al. [34] and Kannan et al. [29] study how a utility-maximizing *decision-maker* may respond to the predictions made by a predictive rule (e.g., the decision-maker may interpret/utilize the predictions a certain way or decide to update the model entirely.) Mouzannar et al. [37] and Heidari et al. [23] propose several dynamics for how individuals within a population may react to predictive rules by changing their qualifications. Dong et al. [15], Hu et al. [26], Milli et al. [36] address *strategic classification*—a setting in which decision subjects are assumed to respond *strategically* and potentially *untruthfully* to the choice of the predictive model, and the goal is to design classifiers that are robust to strategic manipulation. Generalizing strategic classification, Perdomo et al. [38] propose a risk-minimization framework for *performative predictions*, which broadly refers to settings in which the act of making a prediction influences the prediction target. *Incentive-aware learning* [45, 2] is another generalization that, at a high-level, seeks to characterize the conditions under which one can train predictive rules that are robust to training data manipulations.

Two additional topics that are conceptually related to our work but differ in their motivating problems and goals are *adversarial prediction* and *strategy-proof regression*. The adversarial prediction prob-

lem [6, 13] is motivated by settings (e.g., spam detection) in which an adversary actively manipulates data to increase the false-negative rate of the classifier. Adversarial predictions have been modeled and analyzed as a zero-sum game [13] or a Stackelberg competition [6]. Strategyproof/truthful linear regression [14, 12, 9] offers mechanisms for incentivizing agents to report their data truthfully.

As mentioned earlier, many of our modeling choices closely follow Kleinberg and Raghavan [30]. Below, we provide a summary of Kleinberg and Raghavan's results and briefly mention some of the recent contributions following their footsteps. While much of prior work on strategic classification views all feature manipulation as undesirable [15, 26, 36], Kleinberg and Raghavan made a distinction between feature manipulation via *gaming* (investing effort to change observable features in a way that has no positive impact on the quality the principal is trying to measure) and feature manipulation via *improvement* (investing effort in such a way that the underlying characteristics the principal is trying to measure are improved). Their model consists of a single round of interaction between a principal and an agent, and their results establish the optimality and limits of linear assessment rules in incentivizing desired effort profiles. Several papers since then have studied similar settings (see, e.g., Miller et al. [35], Frankel and Kartik [19]) with goals that are distinct from ours. (For example, Frankel and Kartik find a fixed-point assessment rule that improves accuracy by under-utilizing the observable data and flattening the assessment rule.)

Finally, we mention that principle-agent games [33] are classic economic tools to model interactions in which a self-interested entity (the agent) responds to the policy/contract enacted by another (the principal) in ways that are contrary to the principle's intentions. The principal must, therefore, choose his/her strategy accounting for the agent's strategic response. Focusing on linear strategies is a common practice in this literature [24, 7, 17]. For simplicity, we present our analysis for linear assessment rules, but later show that the class of all linear assessment policies is equally as powerful as the class of all monotone assessment policies (Theorem 3.4).

## 2 Problem formulation

In our *stateful* strategic regression setting, a principal interacts with the *same* agent over the course of $T$ time-steps, modeled via a Stackelberg game.[1] The principal moves first, announcing an *assessment policy*, which consists of a *sequence* of assessment rules given by parameters $\{\boldsymbol{\theta}_t\}_{t=1}^{T}$. Each $\boldsymbol{\theta}_t$ is used for evaluating the agent at round $t = 1, \cdots, T$. The agent then best responds to this assessment rule by investing effort in different activities, which in turn produces a series of observable features $\{\mathbf{o}_t\}_{t=1}^{T}$ that maximize his overall score. Through each assessment round $t \in \{1, \cdots, T\}$, the agent receives a score $y_t = f(\boldsymbol{\theta}_t, \mathbf{o}_t)$, where $\boldsymbol{\theta}_t$ is the principal's assessment parameters for round $t$, and $\mathbf{o}_t$ is the agent's observable features at that time. Following Kleinberg and Raghavan, we focus on monotone assessment rules.

**Definition 2.1** (Monotone assessment rules). A assessment rule $f(\boldsymbol{\theta}, \cdot) : \mathbb{R}^n \to \mathbb{R}$ is *monotone* if $f(\boldsymbol{\theta}, \mathbf{o}) \geq f(\boldsymbol{\theta}, \mathbf{o}')$ for $o_k \geq o'_k \; \forall k \in \{1, ..., n\}$. Additionally, $\exists k \in \{1, ..., n\}$ such that strictly increasing $o_k$ strictly increases $f(\boldsymbol{\theta}, \mathbf{o})$.

For convenience, we assume the principal's assessment rules are linear, that is, $y_t = f(\boldsymbol{\theta}_t, \mathbf{o}_t) = \boldsymbol{\theta}_t^\top \mathbf{o}_t$. Later we show that the linearity assumption is without loss of generality. We also restrict $\boldsymbol{\theta}_t$ to lie in the $n$-dimensional probability simplex $\Delta^n$. That is, we require each component of $\boldsymbol{\theta}_t$ to be at least $0$ and the sum of the $n$ components equal $1$.

**From effort investments to observable features and internal states.** The agent can modify his observable features by investing effort in various activities. While these effort investments are private to the agent and the principal cannot directly observe them, they lead to features that the principal can observe. In response to the principal's assessment policy, The agent plays an *effort policy*, consisting of a *sequence* of effort profiles $\{\mathbf{e}_t\}_{t=1}^{T}$ where each individual coordinate of $\mathbf{e}_t$ (denoted by $e_{t,j}$) is a function of the principal's assessment policy $\{\boldsymbol{\theta}_t\}_{t=1}^{T}$. Specifically, the agent chooses his policy $(\mathbf{e}_1, \cdots, \mathbf{e}_T)$, so that it is a best-response to the the principal's assessment policy $(\boldsymbol{\theta}_1, \cdots, \boldsymbol{\theta}_T)$.

Next, we specify how effort investment translates into observable features. We assume an agent's observable features in the first round take the form $\mathbf{o}_1 = \mathbf{o}_0 + \boldsymbol{\sigma}_W(\mathbf{e}_1)$, where $\mathbf{o}_0 \in \mathbb{R}^n$ is the initial value of the agent's observable features *before* any modification, $\mathbf{e}_1 \in \mathbb{R}^d$ is the effort the agent

---

[1] To improve readability, we adopt the convention of referring to the principal as she/her and the agent as he/him throughout the paper.

expends to modify his features in his first round of interaction with the principal, and $\boldsymbol{\sigma}_W : \mathbb{R}^d \to \mathbb{R}^n$ is the *effort conversion function*, parameterized by $W$. The effort conversion function is some concave mapping from effort expended to observable features. (For example, if the observable features in the classroom setting are test and homework scores, expending effort studying will affect both an agent's test and homework scores, although it may require more studying to improve test scores from $90\%$ to $100\%$ than from $50\%$ to $60\%$.)

Over time, effort investment can accumulate. (For example, small businesses accumulate *wealth* over time by following good business practices. Students *learn* as they study and accumulate *knowledge*.) This accumulation takes the form of an internal *state*, which has the form $\mathbf{s}_t = \mathbf{s}_0 + \Omega \sum_{i=1}^{t-1} \mathbf{e}_i$. Here $\Omega \in \mathbb{R}^{d \times d}$ is a *diagonal* matrix in which $\Omega_{j,j}, j \in \{1, \ldots, d\}$ determines how much one unit of effort (e.g., in the $j$th effort coordinate, $e_j$) rolls over from one time step to the next, and $\mathbf{s}_0$ is the agent's initial "internal state". An agent's observable features are, therefore, a function of both the effort he expends, as well as his internal state. Specifically, $\mathbf{o}_t = \boldsymbol{\sigma}_W(\mathbf{s}_t + \mathbf{e}_t)$ (here $\boldsymbol{\sigma}_W(\mathbf{s}_0)$ is analogous to $\mathbf{o}_0$ in the single-shot setting). Note that while for simplicity, we assume the accumulating effort types are socially desirable, our results apply as well to settings where undesirable efforts can similarly accumulate.

**Utility functions for the agent and the principal.** Given the above mapping, the agent's goal is to pick his effort profiles so that the observable features they produce maximize the *sum* of his scores over time, that is, the agent's utility $= \sum_{t=1}^{T} y_t = \sum_{t=1}^{T} \boldsymbol{\theta}_t^\top \mathbf{o}_t$. Our focus on the sum of scores over time is a conventional choice and is motivated by real-world examples. (A small business owner who applies for multiple loans cares about the cumulative amount of loans he/she receives. A student taking a series of exams cares about his/her average score across all of them.)

The principal's goal is to choose his assessment rules over time so as to maximize cumulative effort investments according to her preferences captured by a matrix $\Lambda$. Specifically, the principal's utility $= \left\| \Lambda \sum_{t=1}^{T} \mathbf{e}_t \right\|_1$. The principal's utility can be thought of as a weighted $\ell_1$ norm of the agent's cumulative effort, where $\Lambda \in \mathbb{R}^{d \times d}$ is a *diagonal* matrix where the element $\Lambda_{jj}$ denotes how much the principal wants to incentivize the agent invest in effort component $e_j$.[2]

**Constraints on agent effort.** As was the case in the single-shot setting of Kleinberg and Raghavan, we assume that the agent's choice of effort $\mathbf{e}_t$ at each time $t$ is subject to a fixed budget $B$ (with respect to the $\ell_1$ norm). Without loss of generality, we consider the case where $B = 1$. We explore the consequences of an alternative agent effort formulation – namely a *quadratic cost penalty* – in Appendix G.

**Proposition 2.2.** It is possible to incentivize a wider range of effort profiles by modeling the principal-agent interaction over *multiple* time-steps, compared to a model which only considers one-shot interactions. See Appendix A for an example which illustrates this phenomena.

## 3 Equilibrium characterization

The following optimization problem captures the expression for the agent's best-response to an arbitrary sequence of assessment rules.[3] (Recall that $d$ refers to the dimension of effort vectors ($\mathbf{e}_t$'s), and $n$ refers to the number of observable features, i.e., the dimension of $\mathbf{o}_t$'s.)

The set of agent best-responses to a linear assessment policy, $\{\boldsymbol{\theta}_t\}_{t=1}^{T}$, is given by the following optimization procedure:

$$\{\mathbf{e}_t^*\}_{t=1}^{T} = \arg\max_{\mathbf{e}_1, \ldots, \mathbf{e}_T} \quad \sum_{t=1}^{T} \boldsymbol{\theta}_t^\top \boldsymbol{\sigma}_W \left( \mathbf{s}_0 + \Omega \sum_{i=1}^{t-1} \mathbf{e}_i + \mathbf{e}_t \right), \quad \text{s.t. } e_{t,j} \geq 0, \quad \sum_{j=1}^{d} e_{t,j} \leq 1 \, \forall t, j$$

The goal of the principal is to pick an assessment policy $\{\boldsymbol{\theta}\}_{t=1}^{T}$ in order to maximize the total magnitude of the effort components she cares about, i.e.

---

[2] Note that while we only consider diagonal $\Omega \in \mathbb{R}_+^{d \times d}$, our results readily extend to general $\Omega, \in \mathbb{R}_+^{d \times d}$. By focusing on diagonal matrices we have a one-to-one mapping between state and effort components. Non-diagonal $\Omega$ corresponds to cases where different effort components can contribute to multiple state components.

[3] Throughout this section when it improves readability, we denote the dimension of matrices in their subscript (e.g., $X_{a \times b}$ means $X$ is an $a \times b$ matrix).

$$\{\boldsymbol{\theta}_t^*\}_{t=1}^T = \arg\max_{\boldsymbol{\theta}_1,\ldots,\boldsymbol{\theta}_T} \left\| \Lambda \sum_{t=1}^T \mathbf{e}_t^*(\boldsymbol{\theta}_t,\ldots,\boldsymbol{\theta}_T) \right\|_1, \quad \text{s.t. } \boldsymbol{\theta}_t \in \Delta^n \; \forall t,$$

where we abuse notation by treating $\mathbf{e}_t^*$ as a function of $(\boldsymbol{\theta}_t,\ldots,\boldsymbol{\theta}_T)$. Substituting the agent's optimal effort policy into the above expression, we obtain the following formalization of the principal's assessment policy:

**Proposition 3.1** (Stackelberg Equilibrium). Suppose the principal's strategy space consists of all sequences of linear monotonic assessment rules. The Stackelberg equilibrium of the stateful strategic regression game, $\left(\{\boldsymbol{\theta}_t^*\}_{t=1}^T, \{\mathbf{e}_t^*\}_{t=1}^T\right)$, can be specified as the following bilevel multiobjective optimization problem. As is standard throughout the literature, we assume that the agent breaks ties in favor of the principal. Moving forward, we omit the constraints on the agent and principal action space for brevity.

$$\{\boldsymbol{\theta}_t^*\}_{t=1}^T = \arg\max_{\boldsymbol{\theta}_1,\ldots,\boldsymbol{\theta}_T} \left\| \Lambda \sum_{t=1}^T \mathbf{e}_t^*(\boldsymbol{\theta}_t,\ldots,\boldsymbol{\theta}_T) \right\|_1, \quad \text{s.t. } \{\mathbf{e}_t^*\}_{t=1}^T = \arg\max_{\mathbf{e}_1,\ldots,\mathbf{e}_T} \sum_{t=1}^T \boldsymbol{\theta}_t^{*\top} \boldsymbol{\sigma}_W \left( \mathbf{s}_0 + \Omega \sum_{i=1}^{t-1} \mathbf{e}_i + \mathbf{e}_t \right)$$

## 3.1 Linear assessment policies are optimal

Throughout our formalization of the Stackelberg equilibrium, we have assumed that the principal deploys *linear* assessment rules, when *a priori* it is not obvious why the principal would play assessment rules of this form. We now show that the linear assessment policy assumption is without loss of generality. We start by defining the concept of *incentivizability* for an effort policy, and characterize it through a notion of a *dominated effort policy*.

**Definition 3.2** (Incentivizability). An effort policy $\{\mathbf{e}_t\}_{t=1}^T$ is *incentivizable* if there exists an assessment policy $\{f(\boldsymbol{\theta}_t, \cdot)\}_{t=1}^T$ for which playing $\{\mathbf{e}_t\}_{t=1}^T$ is *a* best response. (Note: $\{\mathbf{e}_t\}_{t=1}^T$ need not be the *only* best response.)

**Definition 3.3** (Dominated Effort Policy). We say the effort policy $\{\mathbf{e}_t\}_{t=1}^T$ is *dominated by* another effort policy if an agent can achieve the same or higher observable feature values by playing another effort policy $\{\mathbf{a}_t\}_{t=1}^T$ that does not spend the full effort budget on at least one time-step.

Note that an effort policy which is dominated by another effort policy will never be played by a rational agent no matter what set of decision rules are deployed by the principal, since a better outcome for the agent will always be achievable.

**Theorem 3.4.** For any effort policy $\{\mathbf{e}_t\}_{t=1}^T$ that is not dominated by another effort policy, there exists a linear assessment policy that can incentivize it.

See Appendix C for the complete proof. We characterize whether an effort *policy* $\{\mathbf{e}_t\}_{t=1}^T$ is dominated or not by a linear program, and show that a *subset* of the dual variables correspond to a linear assessment policy which can incentivize it. Kleinberg and Raghavan present a similar proof for their setting, defining a linear program to characterize whether an effort *profile* $\mathbf{e}_t$ is dominated or not. They then show that if an effort profile is *not* dominated, the dual variables of their linear program correspond to a linear assessment rule which can incentivize it. While the proof idea is similar, their results do not extend to our setting because our linear program must include an additional constraint for every time-step to ensure that the budget constraint is always satisfied. We show that by examining the complementary slackness condition, we can upper-bound the gradient of the agent's cumulative score with respect to a subset of the dual variables $\{\boldsymbol{\lambda}_t\}_{t=1}^T$ (where each upper bound depends on the "extra" term $\gamma_t$ introduced by the linear budget constraint for that time-step). Finally, we show that when an effort policy is not dominated, all of these bounds hold with equality and, because of this, the subset of dual variables $\{\boldsymbol{\lambda}_t\}_{t=1}^T$ satisfy the definition of a linear assessment policy which can incentivize the effort policy $\{\mathbf{e}_t\}_{t=1}^T$.

## 4 Equilibrium computation for linear effort conversions

While the optimization in Proposition 3.1 is nonconvex in general, we provide polynomial-time algorithms for settings in which the agent's effort conversion function can reasonably be viewed as being linear, i.e. $\boldsymbol{\sigma}_W = W$, where $W \in \mathbb{R}^{n \times d}$ is the agent's *effort conversion matrix*. Each component $w_{ij}$ of $W$ is a nonnegative term which represents how much an increase in observable

feature $i$ one unit of effort in action $j$ translates to. While this assumption may not be realistic in some settings, it may work well for others and is a common assumption in the strategic classification literature (e.g., [42, 15, 4]).

**Overview of our solution.** Under settings in which the effort conversion function is linear, we can rewrite the game's Stackelberg Equilibrium in a simplified form (Proposition 4.1). Under this formulation, the agent's optimal effort policy can be computed by solving a *sequence* of linear programs, but computing the principal's optimal assessment policy is a nonconvex optimization problem. However, when we write the principal's objective in terms of the agent's efforts (incentivized by the principal's policy), the function becomes convex. Given this observation, we design an algorithm to optimize the principal's objective over the the set of incentivizable effort profiles (instead of over the principal's policy space). To perform the optimization via convex optimization methods, we first establish that the set of effort profiles is convex and provide a *membership oracle* that determines if an effort profile belongs to this set. Given the membership oracle, we leverage the convex optimization method in Kalai and Vempala [28] to find the (approximate) optimal incentivizable effort profile with high probability. Given this effort policy, we can use the dual of our membership oracle to recover a linear assessment policy which can incentivize it. We begin by characterizing the Stackelberg Equilibrium in this setting.

**Proposition 4.1** (Stackelberg Equilibrium). Suppose the agent's effort conversion function $\boldsymbol{\sigma}_W$ is linear. The Stackelberg equilibrium of the stateful strategic regression game, $\left(\{\boldsymbol{\theta}_t^*\}_{t=1}^T, \{\mathbf{e}_t^*\}_{t=1}^T\right)$, can then be specified as follows:

$$
\begin{aligned}
\forall t : \mathbf{e}_t^* &= \arg\max_{\mathbf{e}_t} \quad \left(\boldsymbol{\theta}_t^{*\top} W + \left(\sum_{i=1}^{T-t} \boldsymbol{\theta}_{t+i}^{*\top}\right) W \Omega\right) \mathbf{e}_t \\
\{\boldsymbol{\theta}_t^*\}_{t=1}^T &= \arg\max_{\boldsymbol{\theta}_1, \dots, \boldsymbol{\theta}_T} \quad \left\| \Lambda \sum_{t=1}^T \arg\max_{\mathbf{e}_t} \left(\boldsymbol{\theta}_t^T W + \sum_{i=1}^{T-t} \boldsymbol{\theta}_{t+i}^\top W \Omega\right) \mathbf{e}_t \right\|_1
\end{aligned}
\tag{1}
$$

*Proof Sketch.* We show that under linear effort conversion functions, the agent's best response problem is *linearly seperable* across time, and the agent's effort profile at each time is given by a linear program. We then plug in each expression for the agent's optimal effort profile at time $t$ into the principal's optimization problem to obtain our final result. See Appendix D for the full proof.

Given the principal's assessment policy $\{\boldsymbol{\theta}_t\}_{t=1}^T$, it is possible to recover the agent's optimal effort policy by solving the linear program for $\mathbf{e}_t$ at each time $t$. On the other hand, recovering the principal's optimal assessment policy is more difficult. The principal's optimal policy takes the form of a multiobjective bilevel optimization problem, a class of problems which are NP-Hard in general [10]. However, we are able to exploit the following proposition to give a polynomial-time algorithm for recovering the principal's optimal assessment policy.

**Proposition 4.2.** The set of incentivizable effort policies is convex if the effort conversion function is linear.

*Proof Sketch.* In order to show that the set of incentivizable effort policies is convex, we assume that it is not and proceed via proof by contradiction. We construct an effort policy $\{\mathbf{z}_t\}_{t=1}^T$ by taking the element-wise average of two incentivizable effort policies $\{\mathbf{x}_t\}_{t=1}^T$ and $\{\mathbf{y}_t\}_{t=1}^T$, and assume it is not incentivizable. Since $\{\mathbf{z}_t^*\}_{t=1}^T$ is not incentivizable, there exists some effort policy $\{\boldsymbol{\zeta}_t^*\}_{t=1}^T$ which dominates it. We show that if this is the case, then $\{\boldsymbol{\zeta}_t^*\}_{t=1}^T$ must dominate either $\{\mathbf{x}_t\}_{t=1}^T$ or $\{\mathbf{y}_t\}_{t=1}^T$. This is a contradiction, since both are incentivizable. See Appendix E.1 for the full proof.

Note that the linear program from Theorem 3.4 can serve as a *membership oracle* for this set. To see this, note that given an effort policy $\{\mathbf{e}_t\}_{t=1}^T$, the LP returns a value of $T$ if and only if $\{\mathbf{e}_t\}_{t=1}^T$ is incentivizable. We now show how to leverage this membership oracle to recover the principal's optimal assessment policy in polynomial time.

Define $\texttt{CvxOracle}(f, \mathcal{M}, R, r, \boldsymbol{\alpha}_0, \epsilon, \delta)$ to be the membership oracle method of Kalai and Vempala [28], which, for a convex set $\mathcal{C}$, takes a linear function $f$ over the convex set $\mathcal{C}$, membership oracle $\mathcal{M}$ to the convex set $\mathcal{C}$, initial point $\boldsymbol{\alpha}_0$ inside of $\mathcal{C}$, radius $R$ of a ball containing $\mathcal{C}$, and a radius $r$ of a ball contained in $\mathcal{C}$ and centered at $\boldsymbol{\alpha}_0$ as input, and returns a member of the convex set which minimizes $f$ up to some $\mathcal{O}(\epsilon)$ term, with probability at least $1 - \delta$. We now present an informal version of their main theorem, followed by our algorithm.

**Theorem 4.3** (Main Theorem of Kalai and Vempala [28] (Informal)). For any convex set $\mathcal{C} \in \mathbb{R}^n$, given a membership oracle $\mathcal{M}$, starting point $\boldsymbol{\alpha}_0$, upper bound $R$ on the radius of the ball containing $\mathcal{C}$, and lower bound $r$ on the radius of the ball containing $\mathcal{C}$, the algorithm of Kalai and Vempala [28] returns a point $\boldsymbol{\alpha}_I$ such that $f(\boldsymbol{\alpha}_I) - \min_{\boldsymbol{\alpha}^* \in \mathcal{C}} f(\boldsymbol{\alpha}^*) \leq \epsilon$ with probability $1 - \delta$, where the number of iterations is $I = \mathcal{O}(\sqrt{n} \log(Rn/r\epsilon\delta))$, and $\mathcal{O}(n^4)$ calls to the membership oracle are made at each iteration.

---

**Algorithm 1:** Assessment Policy Recovery

---

**Result:** An assessment policy $\{\boldsymbol{\theta}_t^*\}_{t=1}^T$

   Define $\mathcal{C}$ to be the set of incentivizable effort policies;

   Let $f(\{\mathbf{e}_t\}_{t=1}^T) = -\left\| \Lambda \sum_{t=1}^T \mathbf{e}_t \right\|_1$, where $\{\mathbf{e}_t\}_{t=1}^T$ is an incentivizable effort policy;

   Define $\mathcal{M}$ to be the linear program from Theorem 3.4;

   Fix an arbitrary assessment policy $\{\boldsymbol{\theta}_{t,0}\}_{t=1}^T$. Solve for initial incentivizable effort policy $\{\mathbf{e}_{t,0}\}_{t=1}^T$ as in Proposition 1;

   Set $R = \sqrt{\frac{T(d-1)}{2(T(d-1)+1)}}$;

   $\{\mathbf{e}_t^*\}_{t=1}^T = \texttt{CvxOracle}(f, \mathcal{M}, R, r, \{\mathbf{e}_{t,0}\}_{t=1}^T, \epsilon, \delta)$;

   Set the primal variables of $\mathcal{M}$ equal to $\{\mathbf{e}_t^*\}_{t=1}^T$, and solve a system of linear equations to recover the dual variables $\{\boldsymbol{\theta}_t^*\}_{t=1}^T$;

---

**Theorem 4.4** (Optimal Assessment Policy). Let $\mathcal{C}$ be the set of incentivizable effort policies. Assuming that $\mathcal{C}$ contains a ball with radius at least $r$ centered at $\{\mathbf{e}_{t,0}\}_{t=1}^T$, the assessment policy $\{\boldsymbol{\theta}_t^*\}_{t=1}^T$ recovered by Algorithm 1 is an $\epsilon$-optimal assessment policy, with probability at least $1 - \delta$.

Before proceeding the proof sketch for Theorem 4.4, we remark that the assumption of $\mathcal{C}$ containing a ball of radius $r$ is commonplace within the membership oracle-based convex optimization literature, both in theory [20, 28], and practice (e.g., [5]). The assumption implies that if it is possible to incentivize an agent to play effort policy $\{\mathbf{e}_{t,0}\}_{t=1}^T$, then it is also possible to incentivize them to play other effort policies within a small margin of $\{\mathbf{e}_{t,0}\}_{t=1}^T$.

*Proof Sketch.* The proof consists of several steps. First, note that the agent's effort policy consists of $T$ $d$-dimensional probability simplexes, which is a $T(d-1)$-dimensional simplex. The circumradius (i.e., the minimum radius of a ball containing the $T(d-1)$-dimensional simplex) is $R = \sqrt{\frac{T(d-1)}{2(T(d-1)+1)}}$. Next, we observe that we can use the linear program defined in the proof of Theorem 3.4 as a membership oracle to the set of incentivizable effort policies. Finally, we observe that the function we are trying to optimize is linear and that it is possible to identify an initial point $\{\mathbf{e}_{t,0}\}_{t=1}^T$ within the convex set $\mathcal{C}$. We can then use a membership oracle-based convex optimization procedure such as Kalai and Vempala [28] to recover the incentivizable effort policy which is most desirable to the principal (up to some $\mathcal{O}(\epsilon)$ term, with high probability) in polynomial time. Given this effort policy, we can use the complementary slackness conditions of our membership oracle linear program to recover the corresponding dual variables, a subset of which will correspond to an assessment policy which can incentivize the agent to play this effort policy. See Appendix E for full details.

The existence of such a membership oracle-based method shows that tractable algorithms exist to recover the principal's optimal assessment policy, and heuristics need not be resorted to under a large class of settings, despite the bilevel multiobjective optimization problem which must be solved.

### 4.1 How many rounds are necessary to implement a desired effort profile?

In the classroom example, we saw that a wider range of effort profiles can be incentivized by extending the fixed budget setting of Kleinberg and Raghavan to multiple time-steps. But how long does the time horizon have to be in order to incentivize a desired effort profile if the principal can pick the time horizon? Additionally, what conditions are sufficient for an effort profile to be incentivizable? We formalize the notion of $(T, t)$-Implementability in the linear effort conversion function setting with these questions in mind.

**Definition 4.5** ($(T, t)$-Implementability). A basis vector $\mathbf{b}_j$ is said to be $(T, t)$-implementable if a rational agent can be motivated to spend their entire effort budget on $\mathbf{b}_j$ for all times $1 \leq t' \leq t$.

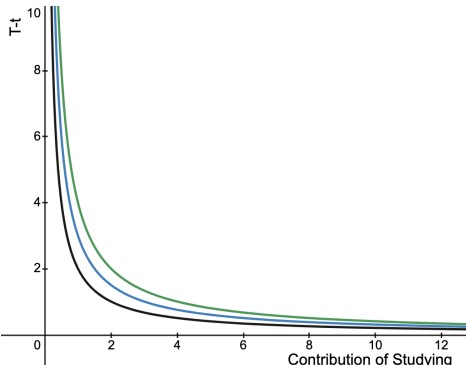

Figure 2: Contribution of studying vs $T - t$. We plot $\Omega_S$ (x-axis) vs $T - t$ (y-axis) for our classroom example in Appendix A. Note that $\Omega_S$ is assumed to be 1 in the appendix. Lighter colors indicate settings in which the student has more incentive to cheat. As long as $\Omega_S > 0$, there exists *some* time horizon under which the student is incentivized to study. As $\Omega_S$ increases, the number of extra time-steps required to incentivize studying decreases.

**Theorem 4.6.** If $T \geq t + \max_c \min_m \frac{\max\{0, W_{mc} - W_{mj}\}}{(\Omega_{jj} W_{mj} - \Omega_{cc} W_{mc})}$ and $\Omega_{jj} W_{mj} - \Omega_{cc} W_{mc} > 0$, then $\mathbf{b}_j$ is $(T, t)$-implementable.

See Appendix F for the full derivation. This bound shows that *any* basis vector is incentivizable if it accumulates faster than other effort profiles. In the worst case, the space of incentivizable effort profiles is the same as in Kleinberg and Raghavan (just set $T = 1$). However, if an effort component accumulates faster than other effort components, there will always exist a time horizon $T$ for which it can be incentivized. In our classroom example, as long as the student retains *some* knowledge from studying, there always will exist a time horizon for which it is possible to incentivize the student to study (see Figure 2). Note that while the principal may be interested in incentivizing more than just basis vectors, there does not appear to be a closed-form lower bound for $T$ for non-basis effort profiles.

## 5 Concluding discussion

We proposed a simple and tractable model in which a principal assesses an agent over a series of timesteps to steer him in the direction of investment in desirable but unobservable types of activities. Our work addresses three crucial gaps in the existing literature, stemming from restricted focus on (1) *short-term* interactions, (2) with *myopic* agents, (3) ignoring the role of earlier effort investments (i.e., the *state*) on future rewards. We observe that within our stateful strategic regression setting, the principal is capable of implementing a more expansive space of average effort investments. Our main results consisted of algorithms for computing the equilibrium of the principal-agent interactions, and characterizing several interesting properties of the equilibrium. There are several natural extensions and directions for future work suggested by our basic model and findings.

**Alternative cost functions.** Following Kleinberg and Raghavan [30], we assumed throughout our analysis that the agent has a *fixed effort budget* in each round. One natural extension of our model is to explore alternative cost formulations for the agent. In Appendix G, we provide the analysis for one natural alternative—that is, a cost term which scales *quadratically* with the total effort expended. Our findings generally remain unaltered. The main qualitative difference between the equilibria of the fixed budget vs. quadratic cost is the following: While under the fixed budget setting, the agent's optimal effort policy is a sequence of basis vectors and the principal's optimal assessment policy generally is not, we find that the opposite is true under the quadratic cost setting. We believe the case-study of quadratic costs provides reassuring evidence for the robustness of our results to the choice of the cost function, however, we leave a more systematic study of equilibrium sensitivity to agent cost function as an interesting direction for future work.

**Bounded rationality.** While we assumed the principal and the agent in our model respond rationally and optimally to each other's strategies, in real-world scenarios, people and institutions are often not fully rational. Therefore, it would be interesting to consider models where our players' rationality is bounded, e.g., by designing assessment policies that are robust to suboptimal effort policies and are capable of implementing desired investments despite the agent's bounded rationality.

**Unknown model parameters & learning.** We assumed the fundamental parameters of our model (e.g., $\boldsymbol{\sigma}_W, \Omega, \Lambda$ and $T$) are public knowledge. It would be interesting to extend our work to settings

where not all these parameters are known. Can we design learning algorithms that allow the players to learn their optimal policy over time as they interact with their counterparts?

**Other simplifying assumptions.** Finally, we made several simplifying assumptions to gain the insights offered by our analysis. In particular, our algorithms for recovering the optimal principal and agent policies relied on the agent having a *linear* effort conversion function. It would be interesting to explore algorithms which work for a wider range of effort conversion functions. Additionally, we assumed that effort expended towards some action was time-independent (e.g., one hour spent studying today is equivalent to one hour spent studying yesterday). It would be interesting to relax this assumption and study settings in which the accumulation of effort is subjected to a discount factor.

## 6   Acknowledgements

This research is supported in part by the NSF FAI Award #1939606. The authors would like to thank Anupam Gupta and Gabriele Farina for helpful discussions about convex optimization techniques, and Gokul Swamy for helpful comments and suggestions.

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
