

(a) Single-step classroom setting.      (b) Multi-step classroom setting.

Figure 3: Comparison between the single-step and multi-step scenarios in the hypothetical classroom setting. The single-step formulation does not account for changes in the student's internal state over time. In the multi-step formulation, effort put towards studying accumulates in the form of knowledge. Modeling this effort accumulation allows the teacher to incentivize the student to study across a wider range of parameter values. The agent can invest effort in 3 actions: cheating on the test (CT), studying (S), and cheating on the homework (CH). $W$ values denote how much one unit of effort translates to the two observable features, test score (T) and homework score (HW). The student's score ($y_t$) at each time-step is a weighted average of these two observable features. In the multi-step setting, $s_t$ denotes the student's internal knowledge state at time $t$.

## A    Formalizing the classroom example

**Example A.1.** We demonstrate this by revisiting the classroom example. Recall that a teacher assigns a student an overall grade $y = \theta_{TE}TE + \theta_{HW}HW$, where $TE$ is the student's test score $HW$ is their homework score, and $\theta_{TE}$ & $\theta_{HW}$ are the weight of each score in the student's overall grade. The student can invest effort into any of three activities: copying answers on the test ($CT$, improves test score), studying ($S$, improves both test and homework score), and looking up homework answers online ($CH$, improves homework score). Suppose the relationship between observable features and effort **e** the agent chooses to spend is defined by the equations

$$TE = TE_0 + W_{CT}CT + W_{ST}S$$

$$HW = HW_0 + W_{SH}S + W_{CH}CH$$

where $TE_0$ and $HW_0$ are the test and homework scores the student would receive if they did not expend any effort. If $W_{CT} = W_{CH} = 3$ and $W_{ST} = W_{SH} = 1$, there is no combination of $\theta_{TE}, \theta_{HW}$ values the teacher can deploy to incentivize the student to study, because the benefit of cheating is just too great. (See [30] for more detail.)

Now consider a multi-step interaction between a teacher and student in which effort invested in studying carries over to future time-steps in the form of knowledge accumulation. The relationships between observable features and effort expended are now defined as

$$TE_t = TE_0 + W_{CT}CT_t + W_{ST}s_t$$

and

$$HW_t = HW_0 + W_{SH}s_t + W_{CH}CH_t$$

where $s_t = \sum_{i=1}^{t} S_i$ is the agent's internal *knowledge state*. Instead of assigning students a single score $y_1$, the teacher assigns the student a score $y_t$ at each round by picking $\theta_{t,T}, \theta_{t,HW}$ at every time-step. The student's grade is then the summation of all scores across time. Suppose $T \geq 3$, where $T$ is the number of rounds of interaction. Consider $W_{CT} = W_{CH} = 3$, $W_{ST} = W_{SH} = 1$, and $TE_0 = HW_0 = 0$. Unlike in the single-round setting, it is easy to verify that students can now be incentivized to study by picking $\theta_{t,TE} = \theta_{t,HW} = 0.5 \ \forall t$.

# B Equilibrium derivations

## B.1 Agent's best-response effort sequence

A rational agent solves the following optimization to determine his best-response effort policy:

$$\{\mathbf{e}_t^*\}_{t=1}^T = \arg\max_{\mathbf{e}_1,\ldots,\mathbf{e}_T} \quad \sum_{t=1}^T (y_t = f_t(\mathbf{e}_1,\ldots,\mathbf{e}_t))$$

$$\text{s.t.} \quad e_{t,j} \geq 0 \; \forall t, j, \quad \sum_{j=1}^d e_{t,j} \leq 1 \; \forall t$$

Recall that the agent's score $y_t$ at each time-step is a function of $(\mathbf{e}_1,\ldots,\mathbf{e}_t)$, the sequence of effort expended by the agent so far. Replacing the score $y_t$ and observable features $\mathbf{o}_t$ with their respective equations, we obtain the expression

$$\{\mathbf{e}_t^*\}_{t=1}^T = \arg\max_{\mathbf{e}_1,\ldots,\mathbf{e}_T} \quad \sum_{t=1}^T \boldsymbol{\theta}_t^\top \boldsymbol{\sigma}_W (\mathbf{s}_t + \mathbf{e}_t)$$

$$\text{s.t.} \quad e_{t,j} \geq 0 \; \forall t, j, \quad \sum_{j=1}^d e_{t,j} \leq 1 \; \forall t$$

where the agent's internal state $\mathbf{s}_t$ at time $t$ is a function of the effort he expends from time 1 to time $t-1$. Replacing $\mathbf{s}_t$ with the expression for agent state, we get

$$\{\mathbf{e}_t^*\}_{t=1}^T = \arg\max_{\mathbf{e}_1,\ldots,\mathbf{e}_T} \quad \sum_{t=1}^T \boldsymbol{\theta}_t^\top \boldsymbol{\sigma}_W \left(\mathbf{s}_0 + \Omega \sum_{i=1}^{t-1} \mathbf{e}_i + \mathbf{e}_t\right)$$

$$\text{s.t.} \quad e_{t,j} \geq 0 \; \forall t, j, \quad \sum_{j=1}^d e_{t,j} \leq 1 \; \forall t$$

# C Proof of Theorem 3.4

*Proof.* Let $\kappa$ be the optimal value of the following linear program:

$$V(\{\mathbf{e}_t\}_{t=1}^T) = \min_{\mathbf{a}_1,\mathbf{a}_2,\ldots,\mathbf{a}_T} \quad \sum_{t=1}^T \|\mathbf{a}_t\|_1$$

$$\text{s.t.} \quad W\left(\Omega \sum_{i=1}^{t-1} \mathbf{a}_i + \mathbf{a}_t\right) \geq W\left(\Omega \sum_{i=1}^{t-1} \mathbf{e}_i + \mathbf{e}_t\right), \; \mathbf{a}_t \geq \mathbf{0}_d, \; \|\mathbf{a}_t\|_1 \leq 1, \; \forall t \tag{2}$$

Optimization 2 can be thought of as trying to minimize the total effort $\{\mathbf{a}_t\}_{t=1}^T$ the agent spends across all $T$ time-steps, while achieving the same or greater feature values at every time $t$ compared to $\{\mathbf{e}_t\}_{t=1}^T$. Let $\{\mathbf{a}_t^*\}_{t=1}^T$ denote the set of optimal effort profiles for Optimization 2. If $\{\mathbf{e}_t\}_{t=1}^T \in \{\mathbf{a}_t^*\}_{t=1}^T$, a value of $\kappa = T$ is obtained. A dominated effort policy is formally defined as follows:

**Lemma C.1** (Dominated Effort Policy). An effort policy $\{\mathbf{e}_t\}_{t=1}^T$ is *dominated by* another effort policy if $\kappa < T$.

The Lagrangian of Optimization 2 can be written as

$$L = \sum_{t=1}^T \|\mathbf{a}_t\|_1 + \sum_{t=1}^T \boldsymbol{\lambda}_t^\top W \left(\Omega \sum_{i=1}^{t-1} (\mathbf{e}_i - \mathbf{a}_i) + \mathbf{e}_t - \mathbf{a}_t\right) + \gamma_t (\|\mathbf{a}_t\|_1 - 1) - \boldsymbol{\mu}_t^\top \mathbf{a}_t,$$

where $\quad \boldsymbol{\lambda}_t \geq \mathbf{0}_n, \; \boldsymbol{\mu}_t \geq \mathbf{0}_d, \; \forall t$

In order for stationarity to hold, $\nabla_{\mathbf{a}_t} L(\mathbf{a}^*, \boldsymbol{\lambda}^*, \boldsymbol{\mu}^*, \boldsymbol{\gamma}^*) = \mathbf{0}_d \ \forall t$, where $\mathbf{x}^*$ denotes the optimal values for variable $\mathbf{x}$. Applying the stationarity condition to Lagrangian function, we obtain

$$\mathbf{1}_d - W^\top \boldsymbol{\lambda}_t^* - \sum_{i=t+1}^T \Omega^\top W^\top \boldsymbol{\lambda}_i^* + \gamma_t^* \cdot \mathbf{1}_d - \boldsymbol{\mu}_t^* = \mathbf{0}_d, \ \forall t \qquad (3)$$

Because of dual feasibility, $\boldsymbol{\mu}_t \geq \mathbf{0}_d \ \forall t$. By rearranging Equation 3 and using this fact, we can obtain the following bound on $W^\top \boldsymbol{\lambda}_t^* + \sum_{t=i+1}^T \Omega^\top W^\top \boldsymbol{\lambda}_t^*$:

$$W^\top \boldsymbol{\lambda}_t^* + \sum_{i=t+1}^T \Omega^\top W^\top \boldsymbol{\lambda}_i^* \leq (1 + \gamma_t^*) \cdot \mathbf{1}_d, \ \forall t \qquad (4)$$

Next we look at the complementary slackness condition. For complementary slackness to hold, $\boldsymbol{\mu}_t^{*\top} \mathbf{a}_t^* = 0 \ \forall t$. If $\kappa = T$, then $\{\mathbf{e}_t\}_{t=1}^T \in \{\mathbf{a}_t^*\}_{t=1}^T$ and therefore $\{\mathbf{e}_t\}_{t=1}^T$ is not dominated. If $\{\mathbf{e}_t\}_{t=1}^T$ is not dominated, $\boldsymbol{\mu}_t^{*\top} \mathbf{e}_t = 0 \ \forall t$. This means that if $e_{t,j} > 0$, $\mu_{t,j} = 0$, $\forall t, j$. This, along with Equation 3, implies that

$$\left[ W^\top \boldsymbol{\lambda}_t^* + \sum_{i=t+1}^T \Omega^\top W^\top \boldsymbol{\lambda}_i^* \right]_j = 1 + \gamma_t^*$$

for all $t, j$ where $e_{t,j} > 0$.

Switching gears, consider the set of *linear* assessment policies $\mathcal{L}$ for which $\{\mathbf{e}_t\}_{t=1}^T$ is incentivizable. The set of linear assessment policies for which $\{\mathbf{e}_t\}_{t=1}^T$ is incentivizable is the set of linear assessment policies for which the derivative of the total score with respect to the agent's effort policy is maximal at the coordinates which $\{\mathbf{e}_t\}_{t=1}^T$ has support on. Denote this set of coordinates as $S$, and the set of coordinates which $\mathbf{e}_t$ has support on as $S_t$. Formally,

$$\mathcal{L} = \left\{ \{\boldsymbol{\theta}_t\}_{t=1}^T \ \middle| \ \left[ \nabla_{\mathbf{a}_t} \sum_{i=1}^T \left( y_i = f\left( \{\mathbf{a}_t\}_{t=1}^T, \{\boldsymbol{\theta}_t\}_{t=1}^T \right) \right) \right]_{S_t} = \max_j \left( \nabla_{\mathbf{a}_t} \sum_{i=1}^T y_i \right) \cdot \mathbf{1}_{|S_t|}, \ \forall t \right\}$$

Recall that $\sum_{t=1}^T y_t = \sum_{t=1}^T \boldsymbol{\theta}_t^\top W \left( \mathbf{s}_0 + \Omega \sum_{i=1}^{t-1} \mathbf{a}_i + \mathbf{a}_t \right)$. Therefore, the gradient of $\sum_{t=1}^T y_t$ with respect to $\mathbf{a}_t$ can be written as

$$\nabla_{\mathbf{a}_t} \sum_{t=1}^T y_t = W^\top \boldsymbol{\theta}_t + \sum_{i=t+1}^T \Omega^\top W^\top \boldsymbol{\theta}_i, \ \forall t$$

Note that the form of $\nabla_{\mathbf{a}_t} \sum_{t=1}^T y_t$ is the same as the LHS of Equation 4. We know that if $\{\mathbf{e}_t\}_{t=1}^T \in \{\mathbf{a}_t^*\}_{t=1}^T$ is incentivizable, the inequality in Equation 4 will hold with equality for all coordinates for which $\{\mathbf{e}_t\}_{t=1}^T$ has positive support. Therefore, the derivative is maximal at those coordinates since it is bounded to be *at most* $1 + \gamma_t^*$, $\forall t$ (due to the KKT conditions for the dominated effort policy linear program). Because of this, $\{\boldsymbol{\lambda}_t^*\}_{t=1}^T$ is in $\mathcal{L}$, which means that $\{\mathbf{e}_t\}_{t=1}^T$ can be incentivized using a linear mechanism.

$\qquad \square$

# D  Equilibrium characterization for fixed budget setting

## D.1  Agent effort policy

**Lemma D.1.** Under linear assessment policy $\{\theta_1, \ldots, \theta_T\}$, a budget constrained agent will play an effort profile from the following set at round $t$:

$$\mathbf{e}_t^* = \arg\max_{\mathbf{e}_t} \quad \left( \boldsymbol{\theta}_t^\top W + \left( \sum_{i=1}^{T-t} \boldsymbol{\theta}_{t+i}^\top \right) W\Omega \right) \mathbf{e}_t$$

$$\text{s.t.} \quad e_{t,j} \geq 0, \ \sum_{j=1}^{T} e_{t,j} \leq 1 \ \forall j$$

*Proof.* The agent's score at each time $y_t$ is a function of $(\mathbf{e}_1, \ldots, \mathbf{e}_t)$. We can replace $y_t$, $\mathbf{o}_t$, and $\mathbf{s}_t$ with their respective equations to get an expression for the agent's optimal effort policy $\{\mathbf{e}_t^*\}_{t=1}^T$ that depends on just $\{\boldsymbol{\theta}_t\}_{t=1}^T$, $s_0$, $W$, and $\Omega$:

$$\{\mathbf{e}_t^*\}_{t=1}^T = \arg\max_{\mathbf{e}_1, \ldots, \mathbf{e}_T} \quad \sum_{t=1}^{T} \boldsymbol{\theta}_t^\top W \left( \mathbf{s}_0 + \Omega \sum_{i=1}^{t-1} \mathbf{e}_i + \mathbf{e}_t \right)$$

$$\text{s.t.} \quad e_{t,j} \geq 0, \ \sum_{j=1}^{T} e_{t,j} \leq 1 \ \forall t, j$$

After expanding the outer sum over the principal assessment rules $\{\boldsymbol{\theta}_t\}_{t=1}^T$, factoring based on the agent's effort at each $t$, and dropping the initial state terms (as they don't depend on $\{\mathbf{e}_1, \ldots, \mathbf{e}_T\}$), we get

$$\{\mathbf{e}_t^*\}_{t=1}^T = \arg\max_{\mathbf{e}_1, \ldots, \mathbf{e}_T} \quad \left( \boldsymbol{\theta}_1^\top W + \left( \sum_{i=1}^{T-1} \boldsymbol{\theta}_{i+1}^\top \right) W\Omega \right) \mathbf{e}_1 + \left( \boldsymbol{\theta}_2^\top W + \left( \sum_{i=1}^{T-2} \boldsymbol{\theta}_{i+2}^\top \right) W\Omega \right) \mathbf{e}_2 + \ldots + \boldsymbol{\theta}_T^\top W \mathbf{e}_T$$

$$\text{s.t.} \quad e_{t,j} \geq 0, \ \sum_{j=1}^{T} e_{t,j} \leq 1 \ \forall t, j \tag{5}$$

Note that the optimization step in (5) is linear in the agent effort policy and can be split into $T$ separate optimization problems, one for each $\mathbf{e}_t$. Thus, the agent can optimize each effort profile $\mathbf{e}_t$ separately by breaking the objective into $T$ parts, each of which is given by the optimization in Lemma D.1. $\qquad \square$

Since the above objective function is linear in $\mathbf{e}_t$, the optimal solution for the agent consists of putting his entire effort budget on the highest-coefficient element of $\boldsymbol{\theta}_t^\top W + \left( \sum_{i=1}^{T-t} \boldsymbol{\theta}_{t+i}^\top \right) W\Omega$. In the classroom setting (Example 1.1), this corresponds to a situation in which the student *only cheats* or *only studies* during each evaluation period. More precisely, let $m$ denote the maximal element(s) of $\boldsymbol{\theta}_t^T W + \sum_{i=1}^{T-t} \boldsymbol{\theta}_{t+i}^\top W\Omega$. We then characterize the set of optimal agent effort profiles at each time-step as $\mathbf{e}_t^* = \mathbb{1}\{j = m\}$ $(1 \leq j \leq d)$. We assume that agents are rational and therefore play an effort policy $\{\mathbf{e}_t\}_{t=1}^T \in \{\mathbf{e}_t^*\}_{t=1}^T$.

## D.2 Principal assessment policy

The goal of the principal is to pick an assessment policy $\{\boldsymbol{\theta}\}_{t=1}^T$ in order to maximize the total magnitude of the agent's cumulative effort in desirable directions (parameterized by $\Lambda$), subject to the constraint that $\boldsymbol{\theta}_t$ lie in the $n$-dimensional probability simplex, i.e.

$$\{\boldsymbol{\theta}_t^*\}_{t=1}^T = \arg\max_{\boldsymbol{\theta}_1,\ldots,\boldsymbol{\theta}_T} \quad \left\|\Lambda \sum_{t=1}^T \mathbf{e}_t(\boldsymbol{\theta}_t,\ldots,\boldsymbol{\theta}_T)\right\|_1 \tag{6}$$
$$\text{s.t.} \quad \boldsymbol{\theta}_t \in \Delta^n \ \forall t$$

From Lemma D.1, we know the form a rational agent's effort $\mathbf{e}_t$ will take for every $t \in \{1,\ldots,T\}$. Substituting this into Equation 6, we obtain the following characterization of the principal's assessment policy:

$$\{\boldsymbol{\theta}_t^*\}_{t=1}^T = \arg\max_{\boldsymbol{\theta}_1,\ldots,\boldsymbol{\theta}_T} \quad \left\|\Lambda \sum_{t=1}^T \arg\max_{\mathbf{e}_t}\left(\boldsymbol{\theta}_t^T W + \sum_{i=1}^{T-t} \boldsymbol{\theta}_{t+i}^\top W\Omega\right)\mathbf{e}_t\right\|_1$$
$$\text{s.t.} \quad \boldsymbol{\theta}_t \in \Delta^n, \ e_{t,j} \geq 0, \ \sum_{j=1}^T e_{t,j} \leq 1 \ \forall t,j$$

# E  Proof of Theorem 4.4

## E.1  The set of incentivizable effort policies is convex

*Proof.* Let the set of incentivizable effort policies be denoted by $I = \left\{\{\mathbf{a}_t\}_{t=1}^T | V\left(\{\mathbf{a}_t\}_{t=1}^T\right) = T\right\}$. In order to show that $I$ is convex, it suffices to show that for all effort policies $\{\mathbf{x}_t\}_{t=1}^T$ and $\{\mathbf{y}_t\}_{t=1}^T \in I$, their element-wise average $\{\mathbf{z}_t\}_{t=1}^T$ also belongs to the set $I$. Let the sets of all possible solutions for for $V(\{\mathbf{x}\}_{t=1}^T)$ and $V(\{\mathbf{y}\}_{t=1}^T)$ be denoted by $\left\{\{\mathbf{e}_{x,t}\}_{t=1}^T\right\} \subseteq I$ and $\left\{\{\mathbf{e}_{y,t}\}_{t=1}^T\right\} \subseteq I$. Since $\{\mathbf{x}_t\}_{t=1}^T \in \left\{\{\mathbf{e}_{x,t}\}_{t=1}^T\right\}$ and $\{\mathbf{y}_t\}_{t=1}^T \in \left\{\{\mathbf{e}_{y,t}\}_{t=1}^T\right\}$, we use $\{\mathbf{x}_t\}_{t=1}^T$ and $\{\mathbf{y}_t\}_{t=1}^T$ as the solutions to $V(\{\mathbf{x}\}_{t=1}^T)$ and $V(\{\mathbf{y}\}_{t=1}^T)$ without loss of generality. Let the agent's observable features at time $t$ when playing effort policy $\{\mathbf{a}_t\}_{t=1}^T$ be denoted by $\mathbf{g}_t(\{\mathbf{a}_t\}_{t=1}^T)$. If $\mathbf{z}_t = \frac{\mathbf{x}_t + \mathbf{y}_t}{2}$ for all $t$, we know that $2\mathbf{g}_t(\{\mathbf{z}_t\}_{t=1}^T) = \mathbf{g}_t(\{\mathbf{x}_t\}_{t=1}^T) + \mathbf{g}_t(\{\mathbf{y}_t\}_{t=1}^T)$ for all $t$, due to the linearity of agent feature values. Moreover, this holds for any combination of effort policies from $\left\{\{\mathbf{e}_{x,t}\}_{t=1}^T\right\}$ and $\left\{\{\mathbf{e}_{y,t}\}_{t=1}^T\right\}$.

Suppose that the effort policy $\{\mathbf{z}\}_{t=1}^T$ is *not* incentivizable. By definition, this must mean that there exists some other effort policy $\{\boldsymbol{\zeta}_t\}_{t=1}^T$ such that an agent can achieve the same feature values at every time-step as he would have received if he had played effort policy $\{\mathbf{z}\}_{t=1}^T$, while expending less total effort at at least one time-step $s$, i.e.

$$\mathbf{g}_t(\{\boldsymbol{\zeta}_t\}_{t=1}^T) = \mathbf{g}_t(\{\mathbf{z}_t\}_{t=1}^T), \ \forall t$$

and

$$\|\boldsymbol{\zeta}_s\|_1 < \|\mathbf{z}_s\|_1, \ s \in \{1,\ldots,T\}.$$

By linearity, $\mathbf{z}_s$'s contribution to the agent's feature values at time $s$ is equal to the average of $\mathbf{x}_s$ and $\mathbf{y}_s$'s contributions to the agent's feature values at time $s$. This means that $2W\boldsymbol{\zeta}_s = 2W\mathbf{z}_s = W\mathbf{x}_s + W\mathbf{y}_s$. Let $\boldsymbol{\zeta}_s^*$ equal $\boldsymbol{\zeta}_s$ rescaled such that $\|\boldsymbol{\zeta}_s^*\|_1 = 1$. $W\boldsymbol{\zeta}_s^* \succcurlyeq W\boldsymbol{\zeta}_s$ and there exists an index $\ell$ such that $[W\boldsymbol{\zeta}_s^*]_\ell > [W\boldsymbol{\zeta}_s]_\ell$ (since we assume the effort conversion matrix $W$ is monotonic). Therefore, $2W\boldsymbol{\zeta}_s^* \succcurlyeq W\mathbf{x}_s + W\mathbf{y}_s$ and $[W\boldsymbol{\zeta}_s^*]_\ell > [W\mathbf{x}_s + W\mathbf{y}_s]_\ell$. Denote the effort policy with the rescaled version of $\boldsymbol{\zeta}_s$ as $\{\boldsymbol{\zeta}^*\}_{t=1}^T = \{\boldsymbol{\zeta}\}_{t=1}^T \backslash \boldsymbol{\zeta}_s \cup \boldsymbol{\zeta}_s^*$. It now follows that $2\mathbf{g}_s(\{\boldsymbol{\zeta}^*\}_{t=1}^T) \succcurlyeq \mathbf{g}_s(\{\mathbf{x}_t\}_{t=1}^T) + \mathbf{g}_s(\{\mathbf{y}_t\}_{t=1}^T)$ and $[2\mathbf{g}_s(\{\boldsymbol{\zeta}^*\}_{t=1}^T)]_\ell > [\mathbf{g}_s(\{\mathbf{x}_t\}_{t=1}^T) + \mathbf{g}_s(\{\mathbf{y}_t\}_{t=1}^T)]_\ell$, which means that $\{\boldsymbol{\zeta}^*\}_{t=1}^T$ must dominate either $\{\mathbf{x}_t\}_{t=1}^T$ or $\{\mathbf{y}_t\}_{t=1}^T$. This is a contradiction, since the effort policies $\{\mathbf{x}_t\}_{t=1}^T$ and $\{\mathbf{y}_t\}_{t=1}^T$ are both incentivizable. Therefore, the set of incentivizable effort policies $I$ must be convex. □

## E.2 Membership oracle-based optimization

Now that we have shown that the set of incentivizable effort policies is convex, we can proceed with our membership oracle-based optimization procedure. Our goal is find the incentivizable effort policy which is most desirable to the principal. Therefore, the function we are trying to minimize is $f(\{\mathbf{a}_t\}_{t=1}^T) = -\|\Lambda \sum_{t=1}^T \mathbf{a}_t\|_1$, where $\{\mathbf{a}_t\}_{t=1}^T$ is an incentivizable effort policy and $\Lambda$ is a diagonal matrix where the element $\Lambda_{jj}$ denotes how much the principal wants to incentivize the agent to invest in effort component $e^{(j)}$. Note that this function is linear, as no element of $\{\mathbf{a}_t\}_{t=1}^T$ can be negative. We also need a membership oracle to the convex set of inventivizable effort policies. Fortunately, Optimization 2 gives us such an oracle. In particular, if a given effort policy $\{\mathbf{e}_t\}_{t=1}^T$ is incentivizable, $V(\{\mathbf{e}_t\}_{t=1}^T)$ will equal $T$. If $\{\mathbf{e}_t\}_{t=1}^T$ is not incentivizable, $V(\{\mathbf{e}_t\}_{t=1}^T)$ will be some value strictly less than $T$. Armed with these tools, all we need is an initial point $\{\mathbf{e}_{t,0}\}_{t=1}^T$ inside the set of incentivizable effort policies to use a membership oracle-based convex optimization procedure such as [28] to recover the agent effort policy which is most desirable to the principal. We can obtain such a point by fixing an arbitrary assessment policy $\{\boldsymbol{\theta}_{t,0}\}_{t=1}^T$ and solving the agent's optimization in Optimization 1 to recover $\{\mathbf{e}_{t,0}\}_{t=1}^T$.

Now that we've found the incentivizable agent effort policy that is (approximatley) most desirable to the principal, we need to find the assessment policy which incentivizes it. Optimization 2 can help us here as well. Recall that if an effort policy $\{\mathbf{e}_t\}_{t=1}^T$ is incentivizable, a subset of the dual variables of Optimization 2 correspond to a linear assessment policy which can incentivize it. So given the incentivizable effort policy which is most desirable to the principal, we can use the complementary slackness conditions of Optimization 2 to recover the assessment policy which can incentivize it.

## F  $(T, t)$-Implementability

*Proof.* From Proposition 3.1, we know that the agent's effort profile $\mathbf{e}_t$ at time $t$ will be a basis vector with weight 1 on the maximal element of $\boldsymbol{\theta}_t^T W + \left(\sum_{i=1}^{T-t} \boldsymbol{\theta}_{t+i}^\top\right) W\Omega$. Therefore, if $\mathbf{b}_j$ is the effort profile induced at time $t$, then

$$\sum_{k=1}^n \left(\theta_{t,k} + \Omega_{jj}\left(\sum_{i=1}^{T-t} \theta_{t+i,k}\right)\right) W_{kj} \geq \sum_{k=1}^n \left(\theta_{t,k} + \Omega_{zz}\left(\sum_{i=1}^{T-t} \theta_{t+i,k}\right)\right) W_{kz}, \text{ for } 1 \leq z \leq d \tag{7}$$

Since we are interested in deriving an upper bound on $T$, we can consider just assessment policies of the form $\boldsymbol{\theta}_t = \boldsymbol{\theta} \; \forall t$ – that is, we limit the principal to employ the same assessment rule across all time-steps. After making this assumption and collecting terms, Equation 7 becomes

$$\sum_{k=1}^n \theta_k \left((W_{kj} - W_{kz}) + (T-t)\left(\Omega_{jj}W_{kj} - \Omega_{zz}W_{kz}\right)\right) \geq 0, \text{ for } 1 \leq z \leq d$$

By solving for $T$, we obtain

$$T \geq t + \frac{\sum_{k=1}^n \theta_k \left(W_{kz} - W_{kj}\right)}{\sum_{k=1}^n \theta_k \left(\Omega_{jj}W_{kj} - \Omega_{zz}W_{kz}\right)}, \text{ for } 1 \leq z \leq d \tag{8}$$

Since the principal employs the same assessment rule across all time-steps, it is optimal for the principal to play $\theta_{t,k} = \mathbb{1}\{k = m\} \; \forall t$, where $m$ is the (non-unique) index of $\boldsymbol{\theta}$ which incentivizes $\mathbf{b}_j$ the most. In other words, $m$ is the index that minimizes the RHS of Equation 8 while still satisfying $\Omega_{jj}W_{kj} \geq \Omega_{zz}W_{kz}$ for all $1 \leq z \leq d$. Equation 8 now becomes

$$T \geq t + \min_k \frac{(W_{kz} - W_{kj})}{(\Omega_{jj}W_{kj} - \Omega_{zz}W_{kz})}, \text{ for } 1 \leq z \leq d \tag{9}$$

Note that if $\Omega_{jj}W_{mj} - \Omega_{zz}W_{mz} \leq 0$ for some $z$, then $\mathbf{b}_j$ will *never* be incentivizable at some generic time $t$, since this means an undesirable effort component accumulates at a rate faster than

effort component $j$. While this claim only holds for static $\boldsymbol{\theta}$-policies, a similar condition holds for the general case - namely the denominator of the bound in Equation 8 must be greater than 0 for all $z$ in order for an effort profile to be incentivizable. In the classroom example, this would correspond to (the somewhat unrealistic) situation in which a student gains knowledge by cheating faster than he does from studying.

Finally, picking the $z$ index which maximizes the RHS of Equation 9 suffices for Equation 9 to hold for $1 \leq z \leq d$. Since $T \geq t$ must hold, the numerator be at least 0.

$$T \geq t + \max_z \min_m \frac{\max\{0, W_{mz} - W_{mj}\}}{(\Omega_{jj} W_{mj} - \Omega_{zz} W_{mz})}$$

$\square$

# G  Alternative agent cost formulation

While we assume that each agent selects their action according to a fixed effort *budget* at every time-step, another common agent cost model within the strategic classification literature is that of a *quadratic cost penalty*. We now explore the use of such a cost formulation in our stateful setting.

## G.1  Agent's best-response effort sequence

Under the quadratic cost setting, a rational agent selects his effort policy in order to maximize his total score minus the quadratic cost of exerting the effort over all time steps. Next, we obtain a close-formed expression for the agent's best-response to an arbitrary sequence of assessment rules under a linear effort conversion function.

**Proposition G.1.** If the effort conversion function has the form $\boldsymbol{\sigma}_W = W$, the set of agent best-responses to a sequence of linear, monotonic assessment rules, $\{\boldsymbol{\theta}_t\}_{t=1}^T$, is $\mathbf{e}_t^* = W^\top \boldsymbol{\theta}_t + (W\Omega)^\top \sum_{i=1}^{T-t} \boldsymbol{\theta}_{t+i} \; \forall t$.

*Proof.* The agent solves the following optimization to determine his best-response effort policy:

$$\{\mathbf{e}_t^*\}_{t=1}^T = \arg\max_{\mathbf{e}_1,\ldots,\mathbf{e}_T} \quad \sum_{t=1}^T (y_t = f_t(\mathbf{e}_1,\ldots,\mathbf{e}_t)) - \frac{1}{2}\|\mathbf{e}_t\|_2^2$$
$$\text{s.t.} \quad e_{t,j} \geq 0 \; \forall t, j$$

Recall that the agent's score $y_t$ at each time-step is a function of $(\mathbf{e}_1, \ldots, \mathbf{e}_t)$, the cumulative effort expended by the agent so far. Replacing the score $y_t$ and observable features $\mathbf{o}_t$ with their respective equations, we obtain the expression

$$\{\mathbf{e}_t^*\}_{t=1}^T = \arg\max_{\mathbf{e}_1,\ldots,\mathbf{e}_T} \quad \sum_{t=1}^T \boldsymbol{\theta}_t^\top W (\mathbf{s}_t + \mathbf{e}_t) - \frac{1}{2}\|\mathbf{e}_t\|_2^2$$
$$\text{s.t.} \quad e_{t,j} \geq 0 \; \forall t, j$$

where the agent's internal state $\mathbf{s}_t$ at time $t$ is a function of the effort he expends from time 1 to time $t-1$. Replacing $\mathbf{s}_t$ with the expression for agent state, we get

$$\{\mathbf{e}_t^*\}_{t=1}^T = \arg\max_{\mathbf{e}_1,\ldots,\mathbf{e}_T} \quad \sum_{t=1}^T \boldsymbol{\theta}_t^\top W \left(\mathbf{s}_0 + \Omega \sum_{i=1}^{t-1} \mathbf{e}_i + \mathbf{e}_t\right) - \frac{1}{2}\|\mathbf{e}_t\|_2^2$$
$$\text{s.t.} \quad e_{t,j} \geq 0 \; \forall t, j$$

Our goal is to separate the above optimization into $T$ separate optimization problems for computational tractability. As a first step towards this goal, we expand the sum over the principal's assessment policy, obtaining the following form:

$$\{\mathbf{e}_t^*\}_{t=1}^{T} = \arg \max_{\mathbf{e}_1,\ldots,\mathbf{e}_T} \quad \boldsymbol{\theta}_1^\top W (\mathbf{s}_0 + \mathbf{e}_1) + \boldsymbol{\theta}_2^\top W (\mathbf{s}_0 + \Omega \mathbf{e}_1 + \mathbf{e}_2) + \ldots + \boldsymbol{\theta}_T^\top W \left( \mathbf{s}_0 + \Omega \sum_{i=1}^{T-1} \mathbf{e}_i + \mathbf{e}_T \right)$$

$$- \frac{1}{2} \left( \|\mathbf{e}_1\|_2^2 + \|\mathbf{e}_2\|_2^2 + \ldots + \|\mathbf{e}_T\|_2^2 \right)$$

$$\text{s.t.} \quad e_{t,j} \geq 0 \; \forall t, j$$

Next, we factor the above based on $\mathbf{e}_t$'s. Additionally, we drop the $\mathbf{s}_0$ terms, since they do not depend on any $\mathbf{e}_t$.

$$\{\mathbf{e}_t^*\}_{t=1}^{T} = \arg \max_{\mathbf{e}_1,\ldots,\mathbf{e}_T} \quad \left( \boldsymbol{\theta}_1^\top W + \sum_{i=1}^{T-1} \boldsymbol{\theta}_{i+1}^\top \Omega W \right) \mathbf{e}_1 - \frac{1}{2} \|\mathbf{e}_1\|_2^2 + \left( \boldsymbol{\theta}_2^\top W + \sum_{i=1}^{T-2} \boldsymbol{\theta}_{i+2}^\top \Omega W \right) \mathbf{e}_2$$

$$- \frac{1}{2} \|\mathbf{e}_2\|_2^2 + \ldots + \boldsymbol{\theta}_T^\top W \mathbf{e}_t - \frac{1}{2} \|\mathbf{e}_T\|_2^2$$

$$\text{s.t.} \quad a_{t,j} \geq 0 \; \forall t, j$$

(10)

Now Equation 10 can be separated based on agent effort profile at each time step $t$. In particular, for $\mathbf{e}_t$ we have:

$$\mathbf{e}_t^* = \arg \max_{\mathbf{e}_t} \quad \left( \boldsymbol{\theta}_t^\top W + \sum_{i=1}^{T-t} \boldsymbol{\theta}_{t+i}^\top \Omega W \right) \mathbf{e}_t - \frac{1}{2} \|\mathbf{e}_t\|_2^2$$

$$\text{s.t.} \quad e_{t,j} \geq 0 \; \forall j$$

Finally, we can get a closed-form solution for each $\mathbf{e}_t^*$ by taking the gradient with respect to $\mathbf{e}_t$ and setting it equal to $\mathbf{0}_d$. Our final expression for $\mathbf{e}_t^*$ is

$$\mathbf{e}_t^* = W^\top \boldsymbol{\theta}_t + (W\Omega)^\top \sum_{i=1}^{T-t} \boldsymbol{\theta}_{t+i} \tag{11}$$

$\square$

**Corollary G.2.** The set of effort profiles the agent can play as a best-response to some linear assessment policy at each time step $t$ grows as the time horizon $T$ increases.

*Proof.* Fix any time horizon $T$ and time step $t \leq T$, the set of effort profiles the agent can play as a best response is a polytope:

$$S_t(T) = \left\{ W^\top \boldsymbol{\theta}_t + (W\Omega)^\top \sum_{i=1}^{T-t} \boldsymbol{\theta}_{t+i} \mid \theta_t, \theta_{t+1}, \ldots, \theta_T \in \Delta_n \right\}$$

The corollary then follows from the fact that $S_t(T) \subset S_t(T+1)$. $\square$

### G.2  Principal's equilibrium assessment policy

Next, given the form of the agent's best response to an arbitrary assessment policy, we can derive the principal's equilibrium strategy as follows:

**Theorem G.3** (Stackelberg Equilibrium). Suppose the principal's strategy space consists of all sequences of linear monotonic assessment rules. The Stackelberg equilibrium of the stateful strategic regression game, $\left( \{\boldsymbol{\theta}_t^*\}_{t=1}^{T}, \{\mathbf{e}_t^*\}_{t=1}^{T} \right)$, can be specified as follows:

$$\forall t : \mathbf{e}_t^* = W^\top \boldsymbol{\theta}_t^* + (W\Omega)^\top \sum_{i=1}^{T-t} \boldsymbol{\theta}_{t+i}^*$$

$$\boldsymbol{\theta}_t^* = \mathbb{1}\{k = \arg\max \| \Lambda \left( I + (t-1)\Omega^\top \right) W^\top \|_1 \}.$$

*Proof.* Proposition G.1 already calculates the agent's best response an arbitrary assessment policy. It only remains to characterize the principal's best response to the agent.

The principal's goal is to maximize the value of the agent's internal state at time $T$. Writing this as an optimization problem, we have

$$\{\boldsymbol{\theta}_t^*\}_{t=1}^T = \arg \max_{\boldsymbol{\theta}_1,\dots,\boldsymbol{\theta}_T} \quad \left\| \Lambda \sum_{t=1}^T \mathbf{e}_t^*(\boldsymbol{\theta}_t,\dots,\boldsymbol{\theta}_T) \right\|_1 \tag{12}$$
$$\text{s.t.} \quad \boldsymbol{\theta}_t \in \Delta^n \; \forall t$$

The sequence $\{\boldsymbol{\theta}_t^*\}_{t=1}^T$ could correspond to a teacher designing a sequence of *(test, homework)* pairs with different weights in order to maximize a student's knowledge, or a bank designing a sequence of evaluation metrics to determine the amount a loan applicant receives when applying for a sequence of loans over time in order to encourage good business practices.

From Equation 11 we know the form of the effort profile at each time for a rational agent. Substituting this into Equation 12, we obtain

$$\{\boldsymbol{\theta}_t^*\}_{t=1}^T = \arg \max_{\boldsymbol{\theta}_1,\dots,\boldsymbol{\theta}_T} \quad \left\| \Lambda \sum_{t=1}^T \left( W^\top \boldsymbol{\theta}_t + (W\Omega)^\top \sum_{i=1}^{T-t} \boldsymbol{\theta}_{t+i} \right) \right\|_1$$
$$\text{s.t.} \quad \boldsymbol{\theta}_t \in \Delta^n \; \forall t$$

As was the case with the agent's optimal effort policy, we would like to separate the optimization for the principal's optimal assessment policy into $T$ separate optimization problems. The current form can be separated based on $\boldsymbol{\theta}$ because we have closed-form solutions for each $\mathbf{e}_t^*$ ($1 \le t \le T$), which are all linear in the principal's assessment policy $\{\boldsymbol{\theta}_t\}_{t=1}^T$:

$$\{\boldsymbol{\theta}_t^*\}_{t=1}^T = \arg \max_{\boldsymbol{\theta}_1,\dots,\boldsymbol{\theta}_T} \quad \mathbf{1}_d^\top \Lambda W^\top \boldsymbol{\theta}_1 + \mathbf{1}_d^\top \Lambda \left( I + \Omega^\top \right) W^\top \boldsymbol{\theta}_2 + \dots + \mathbf{1}_d^\top \Lambda \left( I + (T-1)\Omega^\top \right) W^\top \boldsymbol{\theta}_T$$
$$\text{s.t.} \quad \boldsymbol{\theta}_t \in \Delta^n \; \forall t$$

We can now solve a separate linear program for each $\boldsymbol{\theta}_t$:

$$\boldsymbol{\theta}_t^* = \arg \max_{\boldsymbol{\theta}_t} \quad \mathbf{1}_d^\top \Lambda \left( I + (t-1)\Omega^\top \right) W^\top \boldsymbol{\theta}_t \tag{13}$$
$$\text{s.t.} \quad \boldsymbol{\theta}_t \in \Delta^n$$

Our final solution for $\boldsymbol{\theta}_t^*$ has the form $\boldsymbol{\theta}_t^* = \mathbb{1}\{k = m\}$, where $m$ denotes the maximal element of $\mathbf{1}_d^\top \Lambda \left( I + (t-1)\Omega^\top \right) W^\top$. $\qquad \square$

### G.3 The dynamicity of equilibrium policies

Given our characterization above, one might wonder if the optimal solution for the principal is to simply play a fixed $\boldsymbol{\theta}$ for all $t \in \{1, \dots, T\}$. We show that this is generally not the case—specifically, due to the role of $t$ in determining the maximal component of vector $\mathbf{1}_d^\top \Lambda \left( I + (t-1)\Omega^\top \right) W^\top$.

**Theorem G.4.** The principal's optimal assessment policy $\{\boldsymbol{\theta}_t^*\}_{t=1}^T$ can contain $n$ distinct assessment rules.

The general idea of the proof is as follows. The optimization problem for principal's assessment rule at each time $t$ (Equation 13) is linear with respect to $t$, so any assessment rule $\boldsymbol{\theta}$ which was optimal at some time $t' < t$ but is no longer optimal at time $t$ will never again be optimal at any time $t'' > t$. (This is because $\mathbf{1}_d^\top \Lambda \left( I + (t-1)\Omega^\top \right) W^\top$ is growing at rate $\mathbf{1}_d^\top \Lambda \Omega^\top W^\top$ with respect to $t$, so an element which was maximal at some time $t'$ but is not maximal anymore must have a smaller rate of change than the current maximal element, and will therefore never be maximal again.) So we can conclude that the number of optimal solutions of Equation 13 is *at most* $n$, since each assessment rule $\boldsymbol{\theta}_t$ in the assessment policy is a basis vector with dimensionality $n$.

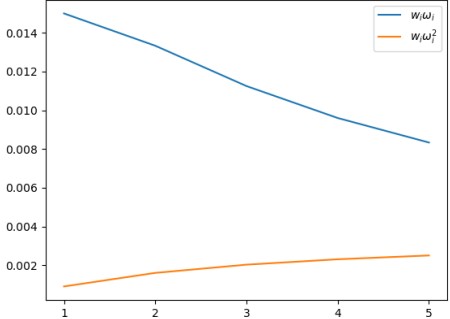 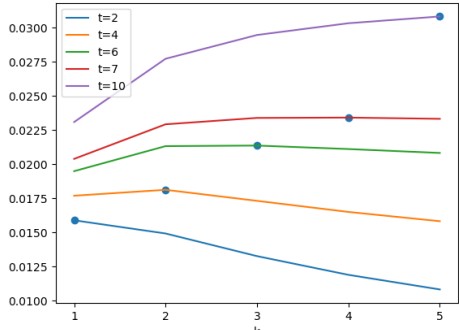

Figure 4: Left: Comparison of the two terms in each component of vector $\mathbf{V}$. The first term decreases as $\frac{1}{k}$, while the second term asymptotically approaches some value as $k$ increases. Right: A scaled version of vector $\mathbf{V}$ evaluated for different values of $t$. The blue circles denote the maximum component of $\mathbf{V}$ for each time $t$. Elements of $\mathbf{V}$ become maximal one-after-another over time.

Next, we provide an example for which there are exactly $n$ optimal solutions. In order to construct such an example, we pick $W$, $\Omega$, and $\Lambda$ to be square, diagonal matrices so that Equation 13 is separable into two terms: one that linearly depends on $t$ and one which has no dependence on $t$. Equation 13 now takes the form $\arg\max_{\boldsymbol{\theta}} \mathbf{V}^{\top}\boldsymbol{\theta}$, where the $k$th element of $\mathbf{V}$ takes the form $W_{kk}\Omega_{kk} + (t-1)W_{kk}\Omega_{kk}^2$. Equation 13 is linear in $\boldsymbol{\theta}$, so $\boldsymbol{\theta}$ will be a basis vector with a 1 at the index where $\mathbf{V}^{\top}$ is maximal and zeros elsewhere. We pick constants $\{W_{kk}\}_{k=1}^{n}$ and $\{\Omega_{kk}\}_{k=1}^{n}$ such that each element $V^{(k)} \in \mathbf{V}$ becomes maximal one-after-one over time. Figure **??** shows how the two terms of $V^{(k)}$ change with $k$. Figure **??** shows how different indices of $\mathbf{V}$ can be maximal for different times.

Next we provide the full proof for the claim that the principal's assessment policy contains $n$ distinct assessment rules.

*Proof.* (Theorem G.4) To show that Equation 13 can have up to $n$ optimal solutions throughout time, it suffices to provide a specific example for which this is the case. Let $\boldsymbol{\theta}, \mathbf{e} \in \mathbb{R}^n$, $\Omega = \Lambda = \in \mathbb{R}^{n \times n}$, and $W = \in \mathbb{R}^{n \times n}$, where $W$ is a diagonal matrix. This corresponds to the case where effort invested in one action corresponds to a change in exactly one observable feature. Under this setting, Equation 13 simplifies to

$$\boldsymbol{\theta}_t = \arg\max_{\boldsymbol{\theta}} \quad [\Omega_{11}W_{11} + (t-1)\,\Omega_{11}^2 W_{11}, \ldots, \Omega_{kk}W_{kk} + (t-1)\,\Omega_{kk}^2 W_{kk}, \ldots,$$
$$\Omega_{nn}W_{nn} + (t-1)\,\Omega_{nn}^2 W_{nn}]^{\top}\boldsymbol{\theta} \tag{14}$$
$$\text{s.t.} \quad \boldsymbol{\theta}_t \in \Delta^n$$

Now let $W_{kk} = \frac{1}{(k+1)^2}$ and $\Omega_{kk} = \frac{k}{100n^3}$ $(1 \leq k \leq n)$. Equation 14 becomes

$$\boldsymbol{\theta}_t = \arg\max_{\boldsymbol{\theta}} \quad \mathbf{V}^{\top}\boldsymbol{\theta}$$
$$\text{s.t.} \quad \boldsymbol{\theta}_t \in \Delta^n \tag{15}$$

where

$$\mathbf{V} =$$
$$\left[ \frac{1}{400n^3}\left(1 + (t-1)\frac{1}{100n^3}\right), \frac{2}{900n^3}\left(1 + (t-1)\frac{2}{100n^3}\right), \ldots, \frac{1}{100n^2(n+1)^2}\left(1 + (t-1)\frac{1}{100n^2}\right) \right]$$

Since Equation 15 is linear in $\boldsymbol{\theta}$, $\boldsymbol{\theta}$ will be a basis vector with support on the element of $\mathbf{V}^\top$ which is maximal. It is therefore sufficient to show that each element of $\mathbf{V}^\top$ is maximal at some point in time. We show via proof by induction that there exists some time $t \in \mathbb{N}$ for which each element of $\mathbf{V}^\top$ is maximal.

**Base case:** $V_1$ is the maximal value of $\mathbf{V}$ when $t = 1$: $\mathbf{V}^\top = \left[ \frac{1}{400n^3}, \frac{2}{900n^3}, \cdots, \frac{1}{100n^2(n+1)^2} \right]^\top$.

**Inductive step:** Assume there is some time $t_k > 1$ such that the $k$th element of $\mathbf{V}$ is maximal. To show that element $k + 1$ is maximal at some time $t_k + \tau_k$ ($\tau_k > 0$), it suffices to show that there exist some $\tau_k$ values such that $V_k < V_{k+1}$ and $V_{k+1} > V_{k+2+m}$ for all $m \geq 0$. It suffices to show this because if $V_k$ is maximal at time $t_k$, $V_{k' < k}$ will never be optimal for times $t_k + \tau_k > t_k$ due to the linearity of the problem.

We first outline the condition for $V_k < V_{k+1}$:

$$\frac{k}{100n^3(k+1)^2}\left(1 + (t_k + \tau_k - 1)\frac{k}{100n^3}\right) < \frac{(k+1)}{100n^3(k+2)^2}\left(1 + (t_k + \tau_k - 1)\frac{k+1}{100n^3}\right)$$

Next we solve for $\tau_k$ and simplify:

$$\tau_k > \frac{100n^3(k^2 + k - 1)}{2k^2 + 4k + 1} - (t_k - 1) \tag{16}$$

We outline a similar condition for $V_{k+1} > V_{k+2+m}$, for all $m \geq 0$:

$$\frac{k+1}{100n^3(k+2)^2}\left(1 + (t_k + \tau_k - 1)\frac{k+1}{100n^3}\right) > \frac{k+2+m}{100n^3(k+3+m)^2}\left(1 + (t_k + \tau_k - 1)\frac{k+2+m}{100n^3}\right)$$

We then solve for $\tau_k$:

$$\tau_k < \frac{100n^3\left((k+1)(k+3+m)^2 - (k+2+m)(k+2)^2\right)}{(k+2+m)^2(k+2)^2 - (k+1)^2(k+3+m)^2} - (t_k - 1) \tag{17}$$

Since Equation 17 needs to hold for *all* $m \geq 0$, it suffices to show that it holds for the value of $m$ which makes the RHS of Equation 17 maximal. To find this $m$ value, we Take the derivative of Equation 17 with respect to $m$ and set it equal to 0. We find that the RHS of Equation 17 is minimized when $m$ is negative. However, $m \geq 0$, so within the constraints of $m$, the RHS of Equation 17 is minimized when $m = 0$. Setting $m = 0$ and simplifying, we obtain

$$\tau_k < \frac{100n^3(k^2 + 3k + 1)}{2k^2 + 8k + 7} - (t_k - 1) \tag{18}$$

We now have sufficient conditions for $V_k < V_{k+1}$ (Equation 16) and $V_{k+1} > V_{k+2+m}$ (Equation 18). Writing the two inequalities together, we see that

$$\frac{k^2 + k - 1}{2k^2 + 4k + 1} < \frac{k^2 + 3k + 1}{2k^2 + 8k + 7}$$

which holds for all values of $k \geq 1$. Therefore, $V_{k+1}$ will be the maximal element of $\mathbf{V}$ at time $t_k + \tau_k$, where

$$\frac{100n^3(k^2 + k - 1)}{2k^2 + 4k + 1} - (t_k - 1) < \tau_k < \frac{100n^3(k^2 + 3k + 1)}{2k^2 + 8k + 7} - (t_k - 1) \tag{19}$$

$\tau_k$ will be strictly greater than 0 for all values of $k$, since $\tau_n > 1$. (This is a sufficient condition for $\tau_k > 0 \; \forall k$ because $\tau_k$ decreases as $k$ increases.) We can see this by subtracting the LHS of Equation 19 from the RHS at $k = n$ to obtain

$$\frac{100n^3 \left(n^2 + 3n + 1\right)}{2n^2 + 8n + 7} - (t_n - 1) - \left(\frac{100n^3 \left(n^2 + n - 1\right)}{2n^2 + 4n + 1} - (t_n - 1)\right) = 200 \frac{n^5 + 4n^4 + 4n^3}{\left(2n^2 + 8n + 7\right) \left(2n^2 + 4n + 1\right)}$$

which is greater than 1 for all values of $n \geq 1$.

Now we characterize a sufficiently long time period for $\mathbf{V}^\top$ to switch to all $n$ values. From Equation 19, we know that

$$T = t_{n-1} + \tau_{n-1} > 1 + \frac{100n^3 \left((n-1)^2 + (n-1) - 1\right)}{2(n-1)^2 + 4(n-1) + 1}$$

Therefore, picking a time horizon such that $T > 100n^3$ is a sufficient condition for the optimal solution of Equation 13 to switch to all $n$ basis vectors. □

### G.4  Optimality of linear assessment policies

So far, for convenience we have focused on *linear* assessment policies for the principal. We next show that this restriction is without loss of generality, that is, linear assessment policies are at least as powerful as the larger class of Lipschitz assessment policies with constant $K \leq 1$, where the comparison is in terms of the effort policies each class can incentivize the agent to play.

**Theorem G.5.** Suppose $K \leq 1$ is constant and $f : \mathbb{R}^n \times \mathbb{R}^n \longrightarrow \mathbb{R}$ is a $K$-Lipschitz function. For any effort policy $\{\mathbf{e}_t\}_{t=1}^T$, if there exists a sequence of assessment rules $\{f(\boldsymbol{\theta}_t', \cdot)\}_{t=1}^T$ to which $\{\mathbf{e}_t\}_{t=1}^T$ is the agent's best-response, then there exists a linear assessment policy $\{\boldsymbol{\theta}_t\}_{t=1}^T$ to which $\{\mathbf{e}_t\}_{t=1}^T$ is also a best-response.

Here is the proof sketch. In order to show that linear assessment policies are optimal, we re-derive the optimal effort policy a rational agent will play for some arbitrary assessment policy $\{f(\boldsymbol{\theta}_t, \cdot)\}_{t=1}^T$. We find that an agent's optimal effort policy is linear in $\{\nabla_{\mathbf{o}_t} f(\boldsymbol{\theta}_t, \cdot)\}_{t=1}^T$, the gradient of the assessment policy with respect to the agent's observable features. Therefore, picking each decision rule to be $f(\boldsymbol{\theta}_t, \mathbf{o}_t) = \boldsymbol{\theta}_t^\top \mathbf{o}_t$ is optimal, assuming no restrictions on $\boldsymbol{\theta}_t$. However, since we restrict each linear decision rule $\theta_t$ to lie in the probability simplex $\Delta^n$, playing the optimal $\{\boldsymbol{\theta}_t\}_{t=1}^T$ is at least as good as any assessment policy in the set of Lipschitz continuous assessment policies with Lipschitz constant $K \leq 1$.

*Proof.* Recall that

$$\{\mathbf{a}_t^*\}_{t=1}^T = \arg \max_{\mathbf{a}_1, \ldots, \mathbf{a}_T} \quad \sum_{t=1}^T y_t - \frac{1}{2}\|\mathbf{a}_t\|_2^2 \tag{20}$$
$$\text{s.t.} \quad a_t^{(j)} \geq 0 \; \forall t, j$$

This is the generic optimization problem for the agent's optimal effort policy $\{\mathbf{a}_t^*\}_{t=1}^T$ from Section G.1. However, instead of specifying the score $y_t$ achieved at each time step to be a linear function of the agent's observable features $\mathbf{o}_t$, we leave the relationship between observable features and score as some generic function $y_t = f(\boldsymbol{\theta}_t, \mathbf{o}_t)$, parameterized by $\boldsymbol{\theta}_t$. We can still obtain an expression for $\mathbf{a}_t^*$ by taking the gradient of Equation 20 with respect to $\mathbf{a}_t$ and setting it equal to $\mathbf{0}_d$. By applying the chain rule, we obtain

$$\mathbf{a}_t^* = \nabla_{\mathbf{a}_t} \sum_{t=1}^T y_t = \sum_{t=1}^T \nabla_{\mathbf{a}_t} f(\boldsymbol{\theta}_t, \mathbf{o}_t) = \sum_{t=1}^T \nabla_{\mathbf{a}_t} W \left(\mathbf{s}_0 + \Omega \sum_{k=1}^{i-1} \mathbf{a_k} + _i\right) \cdot \nabla_{\mathbf{o}_t} f(\boldsymbol{\theta}_t, \mathbf{o}_t)$$

$$\mathbf{a}_t^* = W^\top \nabla_{\mathbf{o}_t} f(\boldsymbol{\theta}_t, \mathbf{o}_t) + \Omega^\top W^\top \sum_{i=t+1}^T \nabla_{\mathbf{o}_i} f(\boldsymbol{\theta}_i, \mathbf{o}_i) \tag{21}$$

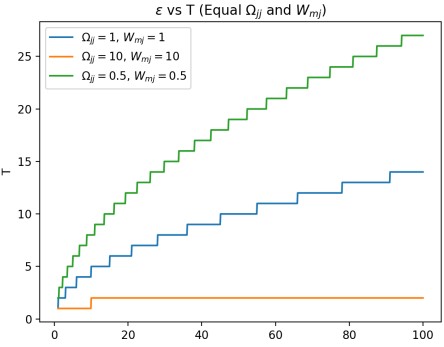
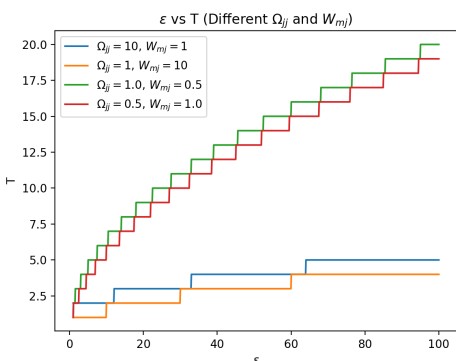

Figure 5: Left: $T$ as a function of $\mathcal{E}$. Larger $\Omega_{jj}$ and $W_{mj}$ terms correspond to fewer time-steps to incentivize $\mathcal{E}$ units of effort. Right: $T$ as a function of $\mathcal{E}$. While $T$ is inversely proportional to both $\Omega_{jj}$ and $W_{mj}$, increasing $\Omega_{jj}$ decreases the time required to incentivize $\mathcal{E}$ units of effort more than an equal increase in $W_{mj}$.

The goal of the principal is to maximize the agent's internal state at time $T$, $\left\| \Lambda \sum_{t=1}^{T} \mathbf{e}_t \right\|_1$. Assuming the agent is rational and plays $\mathbf{e}_t = \mathbf{a}_t^*$, $\forall t$, we can plug Equation 21 into this expression and simplify to obtain

$$\left\| \Lambda \sum_{t=1}^{T} \mathbf{e}_t \right\|_1 = \left\| \Lambda W^\top \sum_{t=1}^{T} \nabla_{\mathbf{o}_t} f(\boldsymbol{\theta}_t, \mathbf{o}_t) \right\|_1 + \left\| \Lambda \Omega^\top W^\top \sum_{t=1}^{T} \sum_{i=t+1}^{T} \nabla_{\mathbf{o}_i} f(\boldsymbol{\theta}_i, \mathbf{o}_i) \right\|_1$$

$$= \left\| \Lambda \sum_{t=1}^{T} \left( I + (t-1)\Omega^\top \right) W^\top \nabla_{\mathbf{o}_t} f(\boldsymbol{\theta}_t, \mathbf{o}_t) \right\|_1$$

Due to the linearity of the problem, the optimal $\nabla_{\mathbf{o}_t} f(\boldsymbol{\theta}_t, \mathbf{o}_t)$ will be basis vectors for all $t$. Since we restrict $\boldsymbol{\theta}_t$ to be in $\Delta^n$, $f(\boldsymbol{\theta}_t, \cdot) = \boldsymbol{\theta}_t$ is at least as optimal as all Lipschitz continuous functions with Lipschitz constant $K \leq 1$. $\qquad \square$

Note that while linear optimality does not hold across the set of *all* assessment policies, this is a result of our restrictions on $\boldsymbol{\theta}_t$ and not due to some suboptimality of linear mechanisms. For example, if we chose to restrict our choice of assessment rules to lie within a probability simplex rescaled by $\Gamma \in \mathbb{R}_+$, then there would exist a linear assessment policy which would be at least as optimal as all Lipschitz functions with Lipschitz constant $K \leq \Gamma$.

### G.5 What levels of effort can be incentivized within $T$ rounds?

According to Corollary G.2, we know that longer time horizons always expand the set of implementable effort sequences. In what follows, we characterize the number of rounds sufficient for reaching a cumulative effort level of $\mathcal{E}$ in a designated effort component.

**Definition G.6** (($T, \mathcal{E}$)-Incentivizability). An effort component $j$ is ($T, \mathcal{E}$)-incentivizable if a rational agent can be motivated to expend at least $\mathcal{E}$ units of effort in the direction of $j$ over $T$ rounds.

**Theorem G.7.** Let $W_{mj}$ denote the maximal element in the $j$th column of $W$. Then if

$$T = \left\lceil \frac{1}{2} - \frac{1}{\Omega_{jj}} + \frac{1}{2}\sqrt{\left(\frac{2}{\Omega_{jj}} - 1\right)^2 + \frac{8\mathcal{E}}{\Omega_{jj}W_{mj}}} \right\rceil, \tag{22}$$

effort component $j$ is ($T, \mathcal{E}$)-incentivizable for $\Omega_{jj} > 0$.

*Proof.* The relationship between total effort $\mathcal{E}$ and the minimum time horizon $T$ required to induce an agent to expend $\mathcal{E}$ units of effort in the direction of effort component $j$ can be written as

$$\min_{\boldsymbol{\theta}_1,\dots,\boldsymbol{\theta}_T} \quad T$$

$$s.t. \quad \mathcal{E} \leq \sum_{t=1}^{T} a_t^{*(j)}, \; \boldsymbol{\theta}_t \in \Delta^n \; \forall t, \; T > 1 \tag{23}$$

where $a_t^{*(j)} = \sum_{k=1}^{n}(\theta_t^{(k)} + \Omega_{jj}(\sum_{i=1}^{T-t} \theta_{t+i}^{(k)}))W_{kj}$ (see Equation 11). Since we only care about the effort accumulated in coordinate $j$ at each time-step, the principal's optimal assessment policy is to pick the assessment rule $\boldsymbol{\theta}_t$ that maximizes the effort the agent expends in coordinate $j$ at time $t$. This translates to picking $\theta_t^{(k)} = \mathbb{1}\{W_{kj} = W_{mj}\} \; \forall t$, where $W_{mk} = \max_k W_{kj}$. In other words, the principal wants to play the same basis vector at every time-step, which will have weight on the observable feature that effort component $j$ contributes the most to. Plugging in this expression for $\theta_t^{(k)}$, the constraint in Equation 23 simplifies to

$$\mathcal{E} \leq \sum_{t=1}^{T} \left(1 + \Omega_{jj}(T-t)\right) W_{mj} = \left(T + \frac{\Omega_{jj}}{2}\left(T^2 - T\right)\right) W_{mj}$$

Note that this will hold with equality if $\mathcal{E} = \sum_{t=1}^{T} a_t^{*(j)}$. After solving for $T$ and simplifying, we get

$$T \geq \frac{1}{2} - \frac{1}{\Omega_{jj}} + \frac{1}{2}\sqrt{\left(\frac{2}{\Omega_{jj}} - 1\right)^2 + \frac{8\mathcal{E}}{\Omega_{jj}W_{mj}}} \tag{24}$$

Since we want the time horizon to be as small as possible but need $T$ to be an integer, we take the ceiling of Equation 24 to get our final time horizon value. $\qquad\square$

Note that the time horizon $T$ scales as $\sqrt{\mathcal{E}}$ because $\mathbf{a}_t^*$, the optimal agent effort profile at time $t$, has a linear dependence on $T - t$, and the total effort $\mathcal{E}$ expended by the agent is proportional to $\sum_{t=1}^{T} a_t^{*(j)}$. Intuitively, this can be seen as the agent choosing to put in most of the work "up front" in order to reap the benefits of his effort across a longer period of time.

Note that the bound on $T$ is tight for $(T, \mathcal{E})$ pairs where $\mathcal{E} = \sum_{t=1}^{T} e_t^{(j)}$. For example, let $j = 1$ and $\Omega = W = I \in \mathcal{R}^{2\times 2}$. If we pick $\boldsymbol{\theta}_t = [1 \quad 0]^\top$, then $e_t^{(1)} = 1 + (T - t)$, from which it is straightforward to see that with 2 total time-steps, the cumulative effort in the direction of $j$ will be 3. By setting $\mathcal{E} = 3$ in Equation 22, we get $T \geq 2$, showing that our lower bound on $T$ is indeed tight for this example.

A natural question is if we can recover a similar definition of $(T, \mathcal{E})$-incentivizability if we want to incentivize some arbitrary subset of effort $\mathbf{e}_S$ over time. While we can obtain a bound for incentivizing *one* index $j \in S$ using the above formulation, obtaining a tighter characterization may require playing different assessment rules over time. Determining these optimal assessment rules requires solving an optimization problem, so a closed-form bound for this setting is not easy to obtain.

### G.6 Discussion: comparing the fixed budget and quadratic cost models

While the principal is able to incentivize a wider range of effort profiles under both the fixed budget and quadratic cost setting, there are several differences in the optimal policies recovered in each setting. In the fixed budget setting, the optimal agent effort policy under linear effort conversion function is to play a basis vector at every time-step (see Proposition 4.1), while the principal's optimal decision rules are generally not basis vectors. Somewhat surprisingly, in the quadratic cost setting the roles are exactly reversed. The principal's optimal linear assessment policy is to play a sequence of basis vectors, while the agent's effort policy will generally involve spending effort in different directions at the same time-step. While in settings such as our classroom example it may be desirable to incentivize agents to play basis vectors (e.g. only study), the choice of constraint on agent effort

is problem-specific and should be chosen based on what is most realistic under the specific setting being studied.

Another difference between the two settings is the computational complexity of recovering the optimal linear policies for the principal and agent. In the fixed budget setting, we can recover the agent's optimal effort policy by solving a sequence of linear programs, and we can recover the principal's optimal assessment policy by using a membership oracle-based method. On the other hand, we have a simple closed-form solution for the agent's optimal effort policy and can recover the principal's optimal linear assessment policy by solving a sequence of linear programs under the quadratic cost formulation.