# OpenReview forum: "Stateful Strategic Regression"
_NeurIPS.cc/2021/Conference — NeurIPS 2021 Poster_

### Official Review · Reviewer_xFkk · 2021-07-03

**Rating:** 8
**Confidence:** 4

**Summary:**

The paper discusses the problem of stateful strategic regression, where the basic idea is to model the interaction between a principal and a strategic agent over multiple time steps. Specifically, the principal announces a sequence of assessment policies, and the agent invests effort to play a best response to this sequence, defined as the effort policy that maximizes the cumulative score the agent receives. An important aspect of the model is that the agent’s effort accumulates over time through an “internal state”. For example, studying for an exam today can help a student’s performance in the following days.

**Limitations And Societal Impact:**

The authors have discussed limitations and possible improvements of their work in a discussion section. I've included various suggestions as part of the main review.

**Main Review:**

I really like the motivation and the idea of studying long-term interactions between the principal and the agent. I particularly like the idea of introducing the agent’s internal state which seems like the right way to capture the compounding nature of efforts. The paper is also well-written overall, with a couple of exceptions where I felt that some claims were a bit too informal (I’m providing specific comments below, together with different minor remarks). The main downside for me is that it’s perhaps unclear what the formal difference is between the setup considered in this paper and a one-shot setting where we increase the expressiveness (e.g., dimension) of the scoring functions and the agent efforts. In particular, the setup can still be thought of as consisting of a single time step because all assessment rules are announced at once, and the agent optimizes their response to all these rules at once. So one can think of the sequence of assessment rules as one high-dimensional rule. Similarly, the sequence of efforts can be seen as as one complex action the agent takes. A more interesting setting might be to consider adaptively chosen assessment rules and agents that plan how to invest their efforts non-myopically despite adaptivity. Still, this paper is a good first step toward understanding strategic regression with a longer time horizon.

It would also be nice to give a bit more intuition for why the membership oracle method due to Lee et al. is the “right” approach. Since the principal’s objective is convex, the most natural approach would be to consider some projected-gradient-descent-like method. The Lee et al. algorithm only assumes an evaluation oracle for the objective, while in this paper the objective is explicit and one can even obtain gradients. So why is the very general and conservative model of Lee et al. the right one? Is it hard to project onto the set of incentivizable effort policies?

This could be a matter of taste, but I felt that a couple of statements were a bit too informal. I don’t doubt that they are true, it’s just that they are not really mathematical claims, but rather intuitive explanations of the underlying mathematical statement. For example, I would formalize the meaning of “wider range of effort policies” in Proposition 2.2 or “rational agents being motivated to spend their entire effort budget” in Definition 4.5. (Note also that incentivizable effort policies, used in Proposition 2.2, are defined only later.) Proposition 2.2 also comes without a proof which makes it even harder to recover the exact mathematical statement.

Some other minor comments:
1. Line 199 says “\sigma_W(s_0) is analogous to o_0 in the single-shot setting”. This only seems to be true if \sigma_W is linear (which is only assumed later on), that is o_1 = \sigma_W(s_0 + e_1) = \sigma_W(s_0) + \sigma_W(e_1). It seems to me that the definition of o_1 in line 185 is not consistent with this general-t equation in line 199.
2. In line 213, it’s not stated in which norm B is the budget.
3. In the equation after line 227, you might want to say that you’re abusing notation by treating e_t as a function. Or you might want to separately define a function that goes from thetas to efforts. Also, I think in the argument it should say theta_1,…,theta_T, not theta_t,…,theta_T?
4. Line 275 says linear twice, “under settings in which the linear effort conversion function is linear“.
5. In Proposition 4.2, it’s not entirely obvious what it means for a set of *sequences* of vectors to be convex. You might want to add a definition.
6. I had trouble understanding the paragraph that starts at line 310. First I was confused what the linear program from Theorem 3.4 even is. It seems to refer to the proof of the theorem, but just from the main body it’s unclear what LP returns T and why. Also there are a few typos in this paragraph - {e_t\|, and there is an extra if in line 311 (I think it should be “given an effort policy” instead of “if given an effort policy”).
7. Definition 3.3 (dominated effort policy) should be defined relative to a sequence of thetas, I think this should be clarified.
8. Line 51, assume -> assumes.
9. Line 55, proposed *by*.
10. Line 107 should have a reference ([31]) after Lee et al. (since this is the first time this work is mentioned).
11. In line 203, I’m not sure if there is a typo, but I find it strange how the equation begins with “=“.
12. First equation after line 291, should the thetas be thetas* (i.e. the equilibrium thetas)?
13. The concluding discussion section has a different title capitalization than the other sections (“Discussion” should be “discussion”).

------------------------------------------------------------------------------------------------------------

Thank you for your response. After discussing the paper with the other reviewers I still think this paper makes a nice conceptual contribution and I'll keep my score.

**Time Spent Reviewing:**

8

---

> ### Author Response · Authors · 2021-08-09
> **Reply to Reviewer xFkk**
>
> We would like to thank the reviewer for their detailed comments and suggestions. We will incorporate them into our revision of the paper. We have two specific responses to points raised by the reviewer:
>
> We agree with the reviewer’s characterization that the principal’s assessment policy can be thought of as one high-dimensional rule and the agent’s effort policy can be seen as one complex action the agent takes. However, we stress that the techniques and methods from the single time-step setting do not carry over straightforwardly into the stateful setting. We believe that studying situations in which all decision rules are not announced by the principal ahead of time is an important direction for future work.
>
> The reviewer makes the conjecture that there may be better algorithms that are more suited to our setting than the membership oracle-based convex optimization procedure we consider. This may very well be the case. Our primary objective in Section 4 is to show that although the optimization problem to recover the principal’s optimal assessment policy takes the form of a multiobjective bilevel optimization problem, polynomial-time algorithms exist to recover the optimal policy. We consider further improvements to the runtime of our algorithm as an important future direction to explore.

---

### Official Review · Reviewer_rp4a · 2021-07-13

**Rating:** 5
**Confidence:** 3

**Summary:**

The paper generalizes an existing problem into a multi-step one. The original problem proposed by Kleinberg and Raghavan is formulated as a Stackelberg game played between a principal and an agent. The principal wants the agent to invest efforts in some given acitivities and uses an assessment policy to score the agent's performance. Once the assessment rule is announced, the agent best responds to the rule by choosing the optimal amount of effort to invest on each activity, aiming to maximize the score it receives. The goal of is to find an assessment rule that will incentivize the agent to invest more efforts. In this paper, the agent's effort investment is a multi-step process. Effort investment in a step will have consequences to the scores in subsequent steps. Hence, both the agent and the principal adopt a long-term vision over their strategies. The paper presented an efficient algorithm to compute the best assessment rule for the principal and also analyzed the time it needs to incentivize a desired effort profile.

**Limitations And Societal Impact:**

I don't find any discussion about negative social impacts in the paper. One possible nagative impact the authors could discuss about is that their algorithms might be used by an unethical principal to exploit the agent.

**Main Review:**

The paper is built on an existing model, which is a simple and natural one for the studied scenario. The extension to the multi-step setting is also realistic and well-motivated. The paper is mostly clear.

The paper showcases a novel algorithm for computing the optimal assessment policy for the principal. The algorithm is tailored to the specific problem setting though.

Some detailed comments:

- The assumption that the amount of effort rolls over to the next step is subject to a transformation Omega looks reasonable at first glance, but from the current definition it looks like this transformation is applied only once to each e_i in the entire process, instead of once every step. Is such a formulation more suitable in some applications? And is it crucial for the main results?

- The definition of dominated effort policy (Definition 3.3) reads a bit loose. Is the notion of dominated defined for a pair of effort policies or only for the one that is dominated? And does the condition need to hold for ALL possible assessment policies of the principal, or for some? It looks more correct to me to define this notion as follows:

An effort policy is dominated if it is worse than some other effort policy no matter what assessment policy the principal uses.

The way Theorem 3.4 is stated also raises the same question. In addition, I don't understand why in Definition 3.3 you need the requirement that the other policy that dominates this one should not spend full effort. I don't find this sufficiently explained in the paper.

- Maybe it's better to point out that if there is a tie, the agent will choose a best-response that maximizes the principal's objective, as is implicitly defined by the bilevel optimization problem.

- Section 4.1 needs more details. It's unclear why consideration of implementable vectors is restricted to basis vectors. I'd suggest that the authors expand this section.

- At Line 181, Page 5, it says that e_t is a function of the assessment policy. This might be confusing as later it is defined that e_1 is a real-valued vector.

- A typo on Page 7 (Line 311)? {e_t}

**Time Spent Reviewing:**

5

---

> ### Author Response · Authors · 2021-08-09
> **Reply to Reviewer rp4a**
>
> We thank the reviewer for their review. Please see our responses below:
>
> The reviewer raises a concern about how we choose to model effort as “rolling over” from one time-step to another. Since we consider finite time horizons, we chose to model effort expended at one time-step as equivalent to effort expended at another. Because of this, the contribution of effort $\mathbf{e}_t$ to the agent’s internal state can be represented by $\Omega \mathbf{e}_t$. This assumption is not critical to our results, our methods should be able to handle any setting under which state is a linear combination of efforts. For instance, it would be straightforward to extend our results to a setting under which the contribution of past efforts to the current state is time-discounted. We would be happy to mention this in our discussion of findings and provide a more formal argument in an appendix. If our response does not adequately address the reviewer’s concern, we can elaborate further.
>
> We thank the reviewer for bringing to our attention the potential confusion that Definition 3.3 may cause. We plan on making this point clearer in future revisions. In short, if an effort policy can achieve the same cumulative score while spending strictly less budget than a competing effort policy, a rational agent will prefer this strategy since they could use their leftover budget to further improve their observable features (and therefore their score).
>
> The reviewer points out that Section 4.1 could use more details. We chose to restrict our consideration in this section to basis vectors, as there appears to be no clean, closed-form bound for non-basis vectors. We will include a discussion of this matter in future revisions.
>
> The reviewer also mentions that our algorithm could be used by an unethical principal to exploit an agent. We will mention in our discussion and broader impact that our model is only applicable in settings where there is a clear desirable direction for effort investments from the society’s perspective.We will include a discussion on this point in future revisions.
>
> We hope our responses clarified any questions the reviewer may have had about our methods. If so, we kindly request that the reviewer consider raising their score.

---

### Official Review · Reviewer_bbo6 · 2021-07-14

**Rating:** 6
**Confidence:** 3

**Summary:**

This paper studies a theoretical model of a multi-round assessment game between a principal and an agent. This model is an extension of Kleinberg and Raghavan (2020) to the multi-time-step setting where the efforts of an agent can accumulate benefits from one time step to the next. It studies the equilibrium computation problem under simplifying linearity assumption. Its main finding is that in the multi-step case, the principal can incentivize a wider range of effort profiles than in the single time step case.

**Limitations And Societal Impact:**

I suggest a more extensive review of related work, and not over-stating the claims (in the writing) that compounding helps the principal incentivize better actions.

**Main Review:**

Originality:
This is an original extension of the model in Kleinberg and Raghavan (2020). The multi-time step case of the "incentivizing effort" problem has yet to be studied, to my knowledge.

The paper's discussion of related emphasizes that related work mostly consider short-term implications of strategic interactions, and that considering "multiple rounds of interactions" is a significant point of novelty/departure. I'm not sure that this is an accurate picture of the literature. For example, this paper does not cite and discuss an earlier work on "long term strategic interactions between decision-makers and decision subjects": Liu et al (2020) ["The Disparate Equilibria of Algorithmic Decision Making when Individuals Invest Rationally."]. It also neglects to mention even earlier work in economics (e.g. Coate and Loury 1993, "Will Affirmative Action Policies Eliminate Negative Stereotypes") that have very much studied this exact topic. Instead of making an overly broad statement about the paper's novelty, I would suggest focusing on the paper's actual contributions, which is to illustrate the effects of *compounded effort* under a stateful model. I believe this is where the originality of the paper lies, and this should be reflected more accurately to the reader.

Quality:

The paper is generally technically sound.

However, there are some statements that I find misleading. For example, it claims in lines 111 to 114 that "Crucially, perhaps our most significant finding is that with multiple rounds of assessments at her disposal, the principal is significantly more effective at incentivizing the agent to accumulate effort in her desired direction (as demonstrated in Figure 1 for a simple teacher-student example)." Does this not depend on whether the desired direction of effort compounds according to the model? Conversely, if the model is such that the student gets better at "copying answers" over time, then the principal would find it even harder to incentivize studying.

In general, there should be more discussion of when this effort compounding might backfire. It's clear to me that there can be no free lunch in this scenario -- increasing the time horizon doesn't automatically lead to better outcomes. This is also not what the theorem statements are saying. Instead of making overly optimistic claims about how this helps the principal in the writing, the paper should be more explicit about the downsides, and about the necessary assumptions for gains.

Clarity:

This is a very well-written paper, in terms of overall clarity and organization.

Significance:

The idea of compounded effort is an interesting one. I see this as the main contribution of the paper, and think it is a good point to be made. Other claims about the field's "restricted focus on short-term interactions and myopic agents" are quite overstated. Re: myopic agents and statefulness, there are a number of papers on fair ML and MDPs, such as Algorithms for Fairness in Sequential Decision Making by Min et al (2021). The current paper should do some comparison to show how they are improving on previous work. See also the references listed in the Quality section.

The actual theoretical results are not remarkable, mostly following from Kleinberg and Raghavan (2020) with some additional elementary arguments (such as linear separability), but are stated cleanly, with helpful exposition of the proof ideas.

Re: "when this effort compounding might backfire", I think the paper's contributions are ambivalent. Unless there is some intrinsic connection between what actions are desirable and what actions compound, basically, the theory still suggests that anything could happen. This is a fine contribution, just perhaps not as important or impactful as it claims.






**Time Spent Reviewing:**

4

---

> ### Author Response · Authors · 2021-08-09
> **Reply to Reviewer bbo6**
>
> We thank the reviewer for their review. Please see our responses below:
>
> The reviewer raises the concern that we possibly overstate the novelty of our work. The main novelty of our results is that we are the first to consider the effects of compounding effort investments over multiple rounds of interaction between a principal and a single agent in the strategic classification/regression setting. This changes our analysis and the resulting insights, compared to both the single shot strategic learning setting and the general principal-agent model. We recognize that we are not the first to consider multiple rounds of interaction in general principal-agent models. That was not our claim and we are happy to further clarify this point by expanding our related work section.
>
> Another concern brought up by the reviewer is the potential downsides of considering stateful settings (e.g., “what if cheating accumulates more than studying does?”). While our analysis readily extends to the accumulation of undesirable efforts, throughout the paper we operate under the assumption that efforts which accumulate (e.g., studying) are socially desirable, whereas efforts which don’t (e.g., cheating) are not. While we believe this is a reasonable assumption for many settings, the reviewer raises a valid concern. The reviewer is correct that if the principal is able to pick the time horizon, it may not necessarily be in their best interest to pick $T > 1$. However, we note that if the time horizon is fixed (i.e., determined externally as is the case in our student-teacher example), it will always be in the principal’s best interest to take these multiple interactions into consideration when determining their assessment policy, even if “bad efforts” accumulate more than “good efforts”. We hope our responses clarified any questions the reviewer may have had about the potential downsides of considering stateful settings. If so, we kindly request that the reviewer consider raising their score.

---

> > ### Comment · Reviewer_bbo6 · 2021-08-30
> > **thank you for the response.**
> >
> > Thank you, I have read the response. I think the response partially addressed my feedback. To be clear, I think this paper did not make it clear that compounding might backfire, and that thee exposition hinges on the idea that only the "good" actions accrue benefits over time. I think this is a really important point to emphasize, if the paper gets published. I appreciate the authors acknowledging this concern and hope the authors will fix this in the revision - in light of that I have raised my score.

---

> > > ### Author Response · Authors · 2021-08-30
> > > **Thanks for reply**
> > >
> > > We thank the reviewer for their helpful comments and for raising their score.

---

### Official Review · Reviewer_TnTg · 2021-07-15

**Rating:** 5
**Confidence:** 4

**Summary:**

The paper studies the problem of strategic regression when the designer’s goal is to incentivize the agents’ effort in the long term. The model follows the one in [Kleinberg and Raghavan, 2020] but considers multiple time steps during which an agent’s effort can accumulate over time in the form of an internal state. The paper characterizes and computes the Stackelberg equilibrium. They also make the following observations: (1) the class of linear policies is as strong as the larger class of monotonic policies; (2) the multi-step policies are more effective than the one-step policies.

**Limitations And Societal Impact:**

Yes.

**Main Review:**

The idea that the designer can use multi-step policies is good, and the observation that multi-step policies can be more effective is interesting. But the technical contribution compared to prior work is not sufficient for NeurIPS.

The word "interaction" used in the statement "we investigate interactions spanning multiple time steps" is misleading. If I understand correctly, by the definition of the Stackelberg equilibrium (Proposition 3.1), there’s no real multi-round interaction between the two sides. The designer posts the full sequence of policies at the beginning and cannot decide the policy based on the agent’s observable features in the previous rounds. As a result, the problem is basically a single-step problem with more variables and slightly different budget constraints, which makes the underlying mathematical problem mostly the same as the one in [Kleinberg and Raghavan, 2020]. (If I understand correctly, the only difference is that we have a budget constraint for each of the rounds here.)

The main technical contribution is Theorem 3.4, which proves that the class of linear policies is as strong as the class of monotonic policies. But the main technique used in the proof is just the duality approach in [Kleinberg and Raghavan, 2020]. Though the problem is not entirely the same, the contribution is not strong enough. Proposition 4.1 does not fully characterize the equilibrium, but only shows that the agent’s best response problem is linearly separable, which is not very insightful. Proposition 4.2 is also quite simple and the proposed algorithm is the standard algorithm for solving convex programs with membership oracles.



**Time Spent Reviewing:**

5

---

> ### Author Response · Authors · 2021-08-09
> **Reply to Reviewer TnTg**
>
> We thank the reviewer for their review. Please see our responses below:
>
>
> The reviewer raises the concern that our work is not sufficiently novel compared to prior work (specifically [Kleinberg and Raghavan, 2019]). However, we argue that the model we study provides both new algorithmic challenges and new qualitative insights.
>
> While the equilibrium characterization may look similar to that of [Kleinberg and Raghavan, 2019] at first glance (indeed, their model is a special case of ours when $T=1$), the effect of multiple time-steps makes the analysis significantly more challenging. For example, consider the problem of recovering the principal’s optimal assessment policy as in Section 4. Under the setting studied by [Kleinberg and Raghavan, 2019], the principal’s optimal assessment policy can be computed efficiently by an algorithm that uses exhaustive search. However, the runtime of such an algorithm is exponential in the time horizon, and is, therefore, computationally inefficient in our stateful setting. On the other hand, the algorithm we provide runs in polynomial time.
>
> In terms of qualitative insights, we provide a sufficient time horizon to incentivize certain desired agent behaviors (see Section 4.1). This bound is unique to our work, and has no analogue in the single time-step setting of [Kleinberg and Raghavan, 2019]. Another novelty of our work is that [Kleinberg and Raghavan, 2019] only consider a budget constraint on the agent’s effort. In addition to this budget constraint, we consider a setting under which the agent pays a quadratic cost for modifying his/her features (another common effort constraint formulation in economics).
>
> The reviewer raises a concern that our use of the word “interaction” is misleading in the statement "we investigate interactions spanning multiple time steps." Note that while the actual interaction between principal and agent plays out over multiple time-steps, we can still model this multi-round interaction via a “one-shot” Stackelberg game. However, analyzing this game, in particular characterizing optimal strategies, still requires reasoning about multiple rounds of interaction. We state this point in the beginning of Section 2 (lines 161-166), although we will consider revising our choice of words in order to clarify this point.
>
> The reviewer mentions that Proposition 4.1 does not fully characterize the equilibrium. We would like to point out that Proposition 4.1 actually does fully characterize the equilibrium. If the reviewer is referring to the fact that the principal and agent optimal strategies, as characterized in Prop. 4.1, are not written in closed form, we would like to remind them that this is generally not possible in Stackelberg games. This is one of the key motivations behind our algorithmic contribution (Algorithm 1). If our response does not satisfy the reviewer’s concern, we are happy to elaborate further.
>
> The reviewer also raises the concern that Proposition 4.2 is “quite simple”. While one may argue that this convexity result is expected, the proof is non-trivial and requires reasoning about agent best responses which have no closed form. Additionally, convexity must be established in order for membership oracle-based convex optimization procedures to be used. Similarly, the reviewer argues that we use a standard algorithm for solving convex programs with membership oracles. While this may be true, our novelty is not in the use of such a method, but in our observation that one can use such membership oracle-based optimization procedures to solve a bilevel multiobjective optimization problem (a class of non-convex problems which is NP-Hard to solve in general) in order to recover the principal’s optimal assessment policy.
>
> We hope our responses clarified any questions the reviewer may have had about the novelty of our methods. If so, we kindly request that the reviewer consider raising their score.

---

> > ### Comment · Reviewer_TnTg · 2021-08-29
> > **Thanks for the detailed replies**
> >
> > Thanks for the detailed replies. I've read them carefully and I have no further questions.

---

> > > ### Author Response · Authors · 2021-08-30
> > > **Thanks for reply**
> > >
> > > Thanks for your reply! If all of the reviewer’s questions about the novelty of our work have been answered, we kindly request that they consider raising their score.

---

### Decision · Program_Chairs · 2021-09-27

**Decision:**

Accept (Poster)

**Comment:**

There is broad agreement among the reviewers on what the paper contributes, but the reviewers evaluate it differently, with one quite strongly in favor and the others more on the fence but not opposed to accepting; part of this is how to value the conceptual aspect of the paper which seems to be its main strength.  I'd recommend a weak accept for this one.  There are some specific comments the authors need to address, and generally we encourage them to be open and transparent, in particular highlighting and explaining the assumption that socially undesirable efforts don't accumulate.